# Silent Neighbors, Loud Secrets: Privacy Leakage from Nearby Classes in Unlearned Models

## Abstract

In this paper, we reveal a significant shortcoming in class unlearning evaluations: overlooking the underlying class geometry can cause privacy leakage. We further propose a simple yet effective solution to mitigate this issue. We introduce a membership-inference attack via nearest neighbors (MIA-NN) that uses the probabilities the model assigns to neighboring classes to detect unlearned samples. Our experiments show that existing unlearning methods are vulnerable to MIA-NN across multiple datasets. We then propose a new fine-tuning objective that mitigates this privacy leakage by approximating, for forget-class inputs, the distribution over the remaining classes that a retrained-from-scratch model would produce. To construct this approximation, we estimate inter-class similarity and tilt the target model's distribution accordingly. The resulting Tilted ReWeighting (TRW) distribution serves as the desired distribution during fine-tuning. We also show that across multiple benchmarks, TRW matches or surpasses existing unlearning methods on prior unlearning metrics. More specifically, on CIFAR-10, it reduces the gap with retrained models by $19\%$ and $46\%$ for U-LiRA and MIA-NN scores, accordingly, compared to the SOTA method for each category.

## 1 Introduction

Machine learning models deployed in real-world applications must support the ability to forget specific information upon request. Class unlearning is particularly challenging because retraining large models from scratch can be prohibitively expensive and it is difficult to change a trained model to completely forget about a learned class or concept. However, privacy regulations such as the EU's GDPR (Voigt & Von dem Bussche, 2017) mandate the "right to be forgotten," enforcing model owners to delete personal data upon request. For instance, in facial recognition systems, all images of a particular individual may constitute a class that must be erased if the person revokes consent. Early methods for machine unlearning primarily expedite the retraining of models at the potential costs of model's performance (Bourtoule et al., 2021). These models also have high computational overhead.

Some other machine unlearning approaches rely on exact methods that are accompanied by certain guarantees. For example, Izzo et al. (2021) introduce an efficient method for deleting data from linear models by adjusting the model parameters directly. Thudi et al. (2022) study unlearning dynamics through the lens of stochastic gradient descent and propose *verification error* as a principled measure of forgetting success. These methods often require specific properties, such as convexity or weight-constrained training, which limits their practicality in deep models. A more practical line of research focuses on approximate methods, that often lack theoretical guarantees but are fast and practical for real-world models. Golatkar et al. (2020) employ fine-tuning techniques with carefully crafted perturbations. Jia et al. (2023) demonstrate that model sparsity can facilitate more efficient unlearning, using pruning to simplify the parameter space. Foster et al. (2024) propose SalUn, a saliency-based technique that identifies and perturbs the most influential weights associated with the forgotten class. Recently, research has shifted toward *class unlearning*, where the goal is to remove an entire semantic class. Warnecke et al. (2021) develop a framework to unlearn both features and labels using influence functions, enabling class-level forgetting in convex models. There are

also several existing approximate unlearning methods (e.g., a person, category, or concept) from a model (Poppi et al., 2024; Chen et al., 2023; Kodge et al., 2024; Chang et al., 2024; Chen et al., 2024; Panda et al., 2024).

The setting of class unlearning is different from unlearning random training samples because the unlearned model has to be depleted of any sign of the unlearned class. Failing to consider this difference, we show that all the existing methods are susceptible to our newly designed variant of membership inference attacks (MIAs), which is based on the *nearest neighbor* of the target class (MIA-NN). This vulnerability has remained unnoticed because previously used evaluations and metrics are mainly designed or inspired by prior work on unlearning random subsets of data, and without paying attention to the properties of a model that is retrained from scratch on all the data other than the target class.

We then propose a new training objective that enhances the robustness to MIA-NN, without computational overhead over finetuning-based unlearning methods. When a class is marked for removal, this method first reassigns its predicted probability mass proportionally to the remaining classes, and then tilts this distribution according to inter-class similarities to better approximate the distribution of the models retrained from scratch on the remaining classes. Our proposed method not only exhibits strong robustness against MIA-NN, it also does not fail the evaluations using prior evaluation metrics.

To evaluate our proposed approaches, we conduct comprehensive experiments on MNIST, CIFAR-10, CIFAR-100, and Tiny-ImageNet, using ten state-of-the-art unlearning methods. In addition to common evaluations in prior work MIA (Hayes et al., 2024), we show robustness against stronger MIA (U-LiRA (Hayes et al., 2024)). Additionally, we present the results for robustness against **MIA-NN**, which we specifically designed to reveal residual leakage of existing methods missed by standard evaluations. Our results show that simple output-space interventions effectively approximates the distribution of retrained models on samples from the forget class without significant computational overhead. Our contributions can be summarized as follows:

- **Membership inference attack via nearest neighbors.** We propose a new membership inference attack, MIA-NN, that utilizes the probabilities assigned to the nearest-neighbor class of the forget class to detect whether its samples have been used for training the model. In contrast to prior work, which primarily relies on simple accuracy drop or binary membership prediction used for random forgetting, our metrics offer deeper insights into how and where leakage persists. MIA-NN reveals vulnerabilities in existing baselines under stronger adversarial scrutiny, setting a new benchmark for effective and privacy-preserving unlearning. Notably, MIA-NN does not assume any access to the training data, which is a more restrictive and realistic setting for adversarial attacks.

- **Lightweight unlearning loss modification for output reweighting.** We propose Tilted ReWeighting (TRW), a simple yet effective modification to the objective of fine-tuning that removes the influence of a forgotten class by approximating the distribution of scratch-retrained models. Unlike prior methods, which fail to replicate the fine-grained behavior of the retrained models, TRW is more robust against MIA-NN and other existing attacks. We accomplish this by applying a post-hoc probability reassignment that adjusts the model's decision boundaries for retained classes. The proposed modification achieves superior performance to existing methods while remaining computationally efficient, making it suitable for deployment in real-world systems that demand rapid unlearning.

## 2 RELATED WORK

In this section, we review prior work on machine and class unlearning, highlighting key approaches such as retraining, fine-tuning, pruning, and representation manipulation.

**Machine Unlearning.** Early work on machine unlearning focused on completely retraining models from scratch on the retained data. While accurate, this approach is often impractical

for deep networks (Bourtoule et al., 2021). Researchers have thus developed approximate methods without full retraining. One line of work uses influence functions to estimate parameter changes from removing specific data points (Warnecke et al., 2021), enabling efficient unlearning on entire features or class labels with theoretical guarantees. Another direction is to approximate data deletion with minimal cost. Izzo et al. (2021) propose a projection-based update for linear and logistic regression models that achieves data deletion in $O(d)$ time. Thudi et al. (2022) unroll the SGD training trajectory to quantify unlearning. And Jia et al. (2023) propose to prune a model before unlearning, which significantly closes the gap between approximate and exact unlearning. Another category of techniques scrubs specific knowledge from model weights via targeted fine-tuning. Golatkar et al. (2020) propose to inject noise guided by the fisher information matrix to remove information about the forget set. More recently, Fan et al. (2023) proposed Saliency Unlearning (SalUn), which computes a weight saliency map to identify parameters influential in the forgetting set and adjusts only those weights. Foster et al. (2024) propose selective synaptic damping, which uses fish information to identify and dampen parameters important to the target data. Cha et al. (2024) performs instance-wise unlearing through intentionally misclassifying forget samples while preserving utility via adversarial regularization and weight-importance constraints. Bonato et al. (2024) proposes a model-agnostic, retain-set–free method that moves forget features toward a nearest wrong class and preserves utility via OOD distillation. Kurmanji et al. (2023) frames unlearning as selective teacher–student training—aligning on retain data and repelling on forget data that improves scalability and MIA robustness.

Recent works tackle the problem of removing an entire class's influence while retaining accuracy on other classes. One strategy is to directly adjust the model's decision boundaries. Chen et al. (2023) propose to shift decision boundaries via fine-tuning on relabeled or pseudo-class data. Chang et al. (2024) propose a Neuron Attribution method using layer-wise relevance propagation to find and perturb class-specific activation paths. And Shen et al. (2024) construct counterfactual samples to unlearn the class by aligning it with random noise representations. Other strategies involve teacher-student learning and representation subtraction. Chundawat et al. (2023) train a student model through a competent teacher (full knowledge) and an incompetent one (forgetting class). Tarun et al. (2023) apply a two-step impair-repair fine-tuning method to rapidly forget the class. And Kodge et al. (2024) identify forget and retain spaces via singular value decomposition and subtract shared components to remove class-specific features.

**Membership Inference Attacks.** A membership inference attack (MIA) tests whether a specific example was part of a model's training set by exploiting the fact that overfitted models tend to behave differently on seen (members) vs. unseen points (non-members). The original MIA has a shadow-model attack that queries a target to collect confidence vectors and trains an attack classifier to decide membership (Shokri et al., 2017). Reframing evaluation toward the low-FPR regime, Carlini et al. (2022) builds a per-example likelihood ratio that improves TPR at small FPRs. And Kodge et al. (2024) proposes a low-cost, high-power MIA through a Support Vector Machine. More recently, Zarifzadeh et al. (2024) designs a robust, low-cost statistical test by composing pairwise likelihood ratios against population draws, outperforming prior methods even with very few reference models. Also by adapting LiRA, Hayes et al. (2024) introduces per-example unlearning MIAs, showing that stronger, tailored attacks reveal overestimated privacy in prior evaluations and can even degrade retain-set privacy, urging more rigorous U-MIA testing

## 3 METHODS

To build a precise understanding of unlearning objectives and challenges, we first formalize the problem in section 3.1. In section 3.2, we analyze how a retrained model behaves on forgotten samples—an aspect largely overlooked in prior work—and use this insight to design a stronger variant of MIA in section 3.3 that reveals vulnerabilities in existing unlearning methods. Motivated by this observation, in section 3.4 we introduce a unlearning method that mitigates this shortcoming and enhances the effectiveness of unlearning.

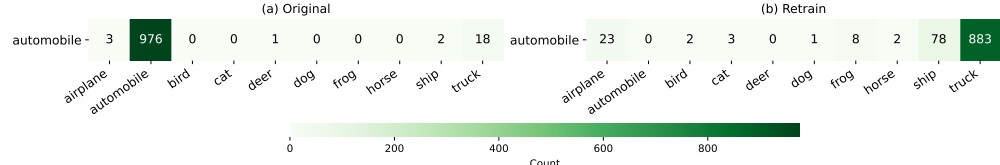

Figure 1: The predicted labels and their corresponding counts for samples belonging to the `automobile` class for the original model (left) and the retrained model (right). Retrain model's predictions on the forget class is skewed toward similar classes.

### 3.1 Problem Definition: Class Unlearning

Let $\mathcal{D} = \{(\mathbf{x}_i, y_i)\}_{i=1}^{N}$ be the full training set with the set of labels $\mathcal{Y}$ containing $K$ different classes. and let $y_f \in \mathcal{Y}$ denote the class to forget. The set of samples to forget is defined as $\mathcal{D}_f = (\mathbf{x}_i, y_i) \subset \mathcal{D} \mid_{y_i = y_f}$, and the retained set is $\mathcal{D}_r = \mathcal{D} \setminus \mathcal{D}_f$. The goal of class unlearning is to produce a model that behaves as if trained only on $\mathcal{D}_r$, i.e., with no influence from $\mathcal{D}_f$.

### 3.2 Motivation

Models retrained on $\mathcal{D}_r$ often assign skewed predictions to $\mathcal{D}_f$ based on semantic similarity. For example, see fig. 1 that shows the prediction on a ResNet18 model trained on CIFAR-10. The *Retrain* model (a model trained on $D_r$) that has never seen `automobile` samples tends to misclassify them as `truck`, which is visually and semantically similar. This behavior is not specific to model architecture but emerges from underlying data-level class similarities. This behavior has been overlooked by prior evaluation methods in class unlearning literature which focus only on the logit or probability for only the forget class.

> **Key Insight:** *The Retrain models exhibit structured misclassifications for forgotten classes—typically to semantically close retained classes. Approximate methods that do not replicate this behavior are susceptible to stronger privacy attacks.*

### 3.3 MIA on the Nearest Neighbor (MIA-NN)

Motivated by the insight in section 3.2, we propose *Membership Inference Attack via Nearest Neighbor* **MIA-NN**, that utilizes the probabilities assigned to the class closest to the forget class. Building on common threat models for unlearning evaluation (Hayes et al., 2024) (see Section A.1), we denote by $\mathcal{M}_o$ the *target* model, by $\mathcal{M}_u$ the *unlearned* model, and by $\mathcal{M}_t$ the *retrained* model. Suppose we have $n$ Retrain models: $\mathcal{M}_t^{[n]} = \{\mathcal{M}_t^{(1)}, \mathcal{M}_t^{(2)}, \dots, \mathcal{M}_t^{(n)}\}$. Note that here we assume access to the training data for training the Retrain models, but we can drop this assumption and derive a fully black-box setting (see section B.9). Let $\mathcal{R} = \{r_1, r_2, \dots, r_{K-1}\}$ be the set of remaining classes, and let $y_f$ denote the forget class. For each class $r_i$, the test set can be split to the following disjoint subsets: $D_{r_i\text{-test}}$ (i.e., test samples belonging to class $r_i$), $D_{f\text{-test}}$ (i.e., test samples belonging to the forget class), and $D_{r_{\hat{i}}\text{-test}}$ where $r_{\hat{i}} = r_j \in \mathcal{R}, j \neq i$ (i.e., the remaining of the test samples).

For a model $\mathcal{M}$ and class $r_i$, let $z_{\mathcal{M}}(x, r_i)$ denote the logit value corresponding to class $r_i$ when sample $x$ is given as input. Now, for each remaining class $r_i$, and for each Retrain model $\mathcal{M}_t^{(j)}$, we construct the training data

$$\mathcal{T}_i^{(j)} = \left\{ \left( z_{\mathcal{M}_t^{(j)}}(x, r_i), 1 \right) \mid x \in D_{r_i\text{-test}} \right\} \cup \left\{ \left( z_{\mathcal{M}_t^{(j)}}(x, r_i), 0 \right) \mid x \in D_{r_{\hat{i}}\text{-test}} \right\}.$$

A binary classifier (e.g., SVM) $h_i^{(j)} : \mathbb{R} \to \{0, 1\}$ is then trained to discriminate $D_{r_i\text{-test}}$ vs. $D_{r_{\hat{i}}\text{-test}}$. The accuracy of this classifier on the forget-class test data is defined as

$$\text{Acc}_i^{(j)} = \frac{1}{|D_{f\text{-test}}|} \sum_{x \in D_{f\text{-test}}} \mathbf{1}\left( h_i^{(j)}(z_{\mathcal{M}_t^{(j)}}(x, r_i)) = 1 \right).$$

For each class $r_i$, we compute the mean accuracy across all the Retrain models to derive: $\bar{\text{Acc}}_i = \frac{1}{n}\sum_{j=1}^{n} \text{Acc}_i^{(j)}$. Finally, we select the *nearest neighbor* class as:

$$r_n \;:=\; \arg\max_{r_i \in \mathcal{R}} \bar{\text{Acc}}_i.$$

To evaluate the unlearned model $\mathcal{M}_u$, we compute

$$\text{Acc}_{r_n}^{\mathcal{M}_u} = \frac{1}{|D_{f\text{-test}}|} \sum_{x \in D_{f\text{-test}}} \mathbf{1}\left(h_{r_n}^{\mathcal{M}_u}(z_{\mathcal{M}_u}(x, r_n)) = 1\right),$$

where $h_{r_n}^{\mathcal{M}_u}$ is trained the same way as the previous classifiers but using logits from $\mathcal{M}_u$ on $D_{r_n\text{-test}}$ and $D_{r_{\hat{n}}\text{-test}}$. We can then consider the gap between $\text{Acc}_{r_n}^{\mathcal{M}_u}$ and $\bar{\text{Acc}}_{r_n}$ as a measure of unlearning effectiveness for model $\mathcal{M}_u$.

Table 1 shows the the value of $\bar{\text{Acc}}_i$ (the `Retrain` column) when forgetting a certain class from a ResNet18 model trained on MNIST, CIFAR-10 and CIFAR-100. As the results show, there is a large gap between $\bar{\text{Acc}}_i$ and $\text{Acc}_{r_n}^{\mathcal{M}_u}$, especially for the more recent unlearning methods that try to optimize for regular MIA score. This reveals a major shortcoming of the current evaluation methods for unlearning and the need for complementary methods such as MIA-NN that perform a more comprehensive analysis of the behavior of unlearned models rather than only focusing on the logit value for the forget class.

| Dataset | Retrain | FT | RL | GA | SalUn | BU | l1 | SVD | SCRUB | SCAR | l2ul | TRW |
|---|---|---|---|---|---|---|---|---|---|---|---|---|
| MNIST (8 → 3) | 74.61 ± 1.2 | 58.67 ± 0.6 | 46.11 ± 0.8 | 19.28 ± 1.6 | 7.39 ± 2.9 | 9.23 ± 3.6 | 5.41 ± 1.8 | 49.56 ± 1.8 | 8.05 ± 2.4 | 44.28 ± 2.3 | 33.41 ± 3.6 | **71.25± 1.7** |
| CIFAR-10 (auto→ truck) | 90.10± 1.0 | 76.65 ± 1.0 | 35.18± 3.4 | 17.14± 2.1 | 6.51± 4.3 | 21.53 ± 1.6 | 10.95 ± 1.9 | 52.23 ± 3.8 | 9.74 ± 1.7 | 56.08± 2.0 | 24.01 ± 1.3 | **83.63± 1.4** |
| CIFAR-100 (beaver → shrew) | 74.87± 1.5 | 56.76 ± 1.0 | 11.08 ± 2.1 | 14.42 ± 1.1 | 6.53 ± 1.2 | 13.73 ± 3.2 | 4.51 ± 3.5 | 39.18 ± 1.1 | 7.54 ± 1.8 | 44.82 ± 1.4 | 19.44 ± 2.7 | **71.42 ± 1.3** |

Table 1: MIA-NN accuracy on the samples from forgotten classes across datasets. Higher values indicate better unlearning (harder to infer membership) based on neighboring classes. The gap with the Retrain models (using three models) reveals under-performance in many of the SOTA unlearning methods that have been evaluated using only regular MIAs.

> **Main Takeaway:** *Existing SOTA methods leak membership under variants of MIAs that probe the fine-grained behavior of the model, such as MIA-NN.*

### 3.4 Tilted ReWeighting (TRW)

As mentioned in prior sections, the susceptibility of existing unlearning methods to MIA-NN arises from the fact that they fail to mimic the fine-grained behavior of the Retrain models on samples from the forget class. A basic solution to enforce this similarity is to utilize the probabilities that the original model assigns to other classes when predicting on the forget samples. That provides us with an estimate of how the probabilities should redistribute when the forget label is enforced to be zero. More specifically, let $p(y \mid \mathbf{x})$ be the original model's output distribution. We perform a rescaling to remove the forget class $y_f$ and derive the reweighted distribution on the remaining classes:

$$\tilde{p}(y \mid x) \;=\; \frac{p(y \mid x)}{1 - p(y_f \mid x)} \quad (y \neq y_f), \qquad \tilde{p}(y_f \mid x) = 0, \tag{1}$$

Using $\tilde{p}$ as the target distribution when fine-tuning on the samples of the forget class, enforces zero probability for the forget label and rescales the probability for other labels to sum up to 1. However, this assumes that the probabilities of other classes would increase proportionally in the absence of the forget class. As figures 3 and 4 in Section A.3 show, this assumption does not hold and the Retrain models will have a much higher bias when predicting on the forget samples. This systematic bias arises from the fact that the forget class has different levels of similarity to other classes, and a model that has never seen the forget class, would assign higher probabilities to more similar classes.

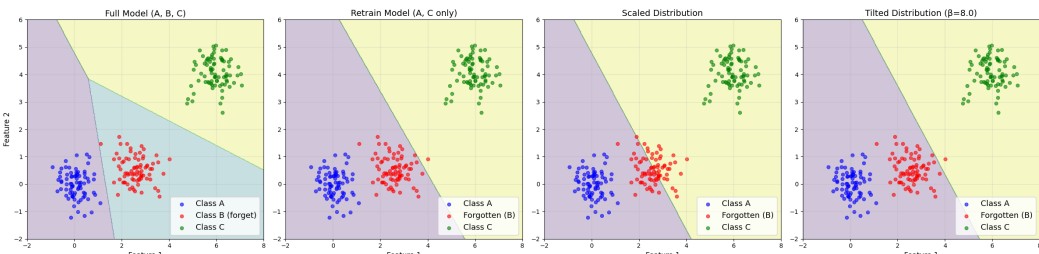

Figure 2: The first figure (from the left) shows the decision boundaries when class B exists. In the Retrain model, $p(y_A|x)/p(y_C|x)$ would mostly increase for $x \in B$ due to the higher similarity of class A and class B (second figure). Third figure shows the decision boundary for a basic rescaling of original model's distribution, while the fourth one shows the tilted distribution, which correctly predicts the decision boundary in the Retrain model.

To capture the systematic bias toward more similar classes, while adding minimal additional constraint to the our target distribution, we impose a first-moment constraint given a set of similarity scores between the forget class and the remaining classes. More specifically, let $s_y$, $y \neq y_f$ show the similarity between class $y$ and $y_f$. Then the set of probability functions over the remaining classes that satisfy this constraint is defined as follows:

$$\mathcal{A} := \Big\{ q \in \Delta^{K-1} : \sum_{y \neq f} q(y \mid x)\, s_y = c \Big\}.$$

Now we need a candidate $q^*(y|x) \in \mathcal{A}$, such that it remains close to the original distribution $p(y|x)$ (and consequently to $\tilde{p}$). In Proposition 3.1 we show that our desired distribution will be a *tilted* version of $\tilde{p}(y|x)$ using the score values and has the following form:

$$q^*(y \mid x) = \frac{\tilde{p}(y \mid x)\, \exp(\beta\, s_y)}{\sum_{j \neq y_f} \tilde{p}(j \mid x)\, \exp(\beta\, s_j)}, \qquad q^*(y_f \mid x) = 0. \tag{2}$$

Note that $\beta \in \mathbb{R}$ controls the strength of the tilt. We do not know the exact value of $c$ a-priori to set it in the constraints, but the hyperparameter $\beta$ helps us achieve various levels of $c$. When $\beta = 0$, the tilted distribution $q^*$ reduces to the plain renormalized $\tilde{p}$, and larger $\beta > 0$ redistributes more of the forget mass to classes similar to $y_f$.

The intuition behind tilted reweighting is shown in Figure 2. The original model has learned decision boundaries to separate class $B$ from the other two classes (first figure from the left). For a model that has been retrained on class $A$ and $C$, the samples from class $B$ will mostly fall within class $A$ and will be assigned to class $A$ with a much higher certainty (second figure). Many of the samples of class $B$ in the original model have a higher probability of being assigned to class $C$ than to class $A$, so a basic rescaling of the conditional probabilities will lead to a decision boundary different from that of the retrained model (third figure). However, by using a tilting function that uses the inverse of the Euclidean distances among the class centroids as the similarity measure, we can recover the decision boundary of the retrained model (code available in the supplementary material).

**Theoretical results.** In Proposition 3.1, we show that equation 2 is equivalent to an *information projection* of the original model's distribution onto the retained simplex under an additional linear constraint on expected similarity (i.e., $\mathcal{A}$). In other words, $q^*$ is the maximum-entropy distribution on the remaining classes that (i) remains close to the baseline $p$ and (ii) while satisfying the bias according to class similarities $s_y$ (see Section A.2 for the proof).

**Proposition 3.1.** *Let $p(\cdot \mid x) \in \Delta^K$ be the distribution of the target model for input $x$, and let $S = \{s_y\}_{y \neq f} \subset \mathbb{R}$ be fixed similarity scores with $y_f$. Given $c \in \mathbb{R}$ in the convex hull of $S$, the information projection of $p$ onto the probability simplex of retrained classes (i.e., $q(\cdot \mid x) \in \Delta^{K-1}$) with linear constraint $\sum_{y \neq f} q(y)\, s_y = c$, has form of equation 2, where $\beta$ is some unique scalar such that $\sum_y q^\star(y \mid x)\, s_y = c$.*

*Remark* 3.2. If $c = \max s_y$ or $c = \min s_y$, the solution concentrates all mass on the maximizing or minimizing classes (corresponding to $\beta^\star = \pm\infty$). Besides, adding a constant shift to all $s_y$ does not change $q^\star$. We set $\beta = 10$ in all experiments to bias the distribution toward more similar classes (see section B.8 for an ablation study on the value of $\beta$).

**Score function.** For the similarity scores we use cosine similarity of the weight vectors corresponding to the logits of each class (see section A.4 for details). We have evaluated other similarity scores based on the distances in the embedding space derived from the target model, but they do not reach the same performance.We leave designing improved similarity functions that better approximate Retrain model behavior to future work.

**Updated loss function.** Now that we have an approximate distribution for how the Retrain model behaves on samples from the forget class, we can utilize it in our loss function for fine-tuning the model. More specifically, we fine-tune the model to minimize the following cross-entropy loss on samples from the forget class:

$$\mathcal{L}_{\text{forget}}(\mathbf{x}) = -\sum_{y=1}^{K} q^*(y \mid \mathbf{x}) \log p(y \mid \mathbf{x}). \tag{3}$$

Therefore, our new objective, Tilted ReWeighting (TRW) loss, can be formulated as:

$$\min_{W} \underbrace{\sum_{(x,y)\in\mathcal{D}_{\text{retain}}} \big[ -\log p(y \mid x) \big]}_{\text{supervised loss on retained-class samples}} + \underbrace{\sum_{x\in\mathcal{D}_f} \big[ \mathcal{L}_{\text{forget}}(x) \big]}_{\text{reweight term on forget samples}} \tag{4}$$

Note that the tilted reweighting loss is a rather general prescription which could be utilized in many of the prior unlearning methods that perform fine-tuning on the target model. For example, some of the prior methods in unlearning focus on sparsification of the parameters that get updated during fine-tuning (Jia et al., 2023; Fan et al., 2023). Although we perform a thorough comparison to these methods, we leave further evaluations on the combination of these methods with TRW to future work.

## 4 Evaluation Setup

In section 4.1, we elaborate on the datasets and model architectures used in our experiments. In section 4.2 the baseline methods are explained, and in section 4.3 we provide the details on the evaluations metrics used for comparison.

### 4.1 Datasets and Models

**Datasets.** To evaluate the effectiveness of our method, we use four image datasets: **MNIST** (Deng, 2012), **CIFAR-10, CIFAR-100** (Krizhevsky et al., 2009), and **TINY-IMAGENET** (Le & Yang, 2015). For single-class forgetting, we report results averaged over different target classes for each dataset. For **MNIST** and **CIFAR-10**, the results are averaged across all 10 classes. For **CIFAR-100**, we compute the average over unlearning experiments conducted on the following 10 randomly selected classes: *apple*, *aquarium fish*, *baby*, *bear*, *beaver*, *bed*, *bee*, *beetle*, *bicycle*, and *bottle*. We use average results on multiple Retrain models as an ideal reference, and assess unlearning methods by how closely they match its performance.

**Models.** For MNIST and CIFAR, we use RESNET18 (He et al., 2016) and VGG19 (Simonyan & Zisserman, 2014) as the original model and for TINY-IMAGENET we use the pretrained RESNET18. Full training and hyperparameter configurations are detailed in section B.2.

### 4.2 Baseline Methods

We compare our method against ten of state-of-the-art machine unlearning baselines, including fine-tuning (FT) (Warnecke et al., 2021), random labeling (RL) (Golatkar et al., 2020), gradient ascent (GA) (Thudi et al., 2022), sparse retraining (l1) (Jia et al., 2023),

boundary unlearning (BU) (Chen et al., 2023), saliency-guided unlearning (SalUn) (Fan et al., 2023), SVD-based feature suppression (SVD) (Kodge et al., 2024), SCRUB (Kurmanji et al., 2023), L2UL (Cha et al., 2024) and SCAR (Bonato et al., 2024). The Retrain models, are considered as the gold standard. Detailed descriptions of all baselines and their hyperparameter configurations are detailed in section B.1 and section B.2, accordingly.

### 4.3 Evaluation Metrics

In section 5.1, following the evaluation metrics used in prior work, we evaluate the unlearned models using three metrics: the accuracy on the remaining set $ACC_r$, the accuracy on the forgetting set $ACC_f$, and the Membership Inference Attack (MIA) score (Shokri et al., 2017). We applied the MIA score used by (Kodge et al., 2024). Ideally, the MIA score of the unlearned model is expected to match that of the retraining model. To perform a comprehensive comparison of the effectiveness of the unlearning methods, we utilized a SOTA MIA called U-LiRA (Hayes et al., 2024) in addition to using the MIAs from prior works. It extends the per-example Likelihood Ratio Attack (LiRA) to the unlearning setting by constructing shadow model distributions that incorporate both the training and unlearning procedures, enabling a more fine-grained, sample-specific membership inference analysis.

## 5 Results

In this section, we present a comprehensive empirical evaluation of our proposed unlearning methods across multiple datasets, architectures, and various evaluation metrics. In section 5.1 we present a thorough comparison using evaluation metrics used in prior unlearning work. In section 5.2 we present the results using a SOTA MIA method that is stronger than prior MIA methods used in unlearning evaluations and can complement the results of MIA-NN for a more comprehensive evaluation. We also evaluate the computation time of our method in section B.6. The ablation study on the value of $\beta$ and other hyper-parameters is presented in section B.8, and the results for unlearning multiple classes can be found in section B.10.

### 5.1 Comparison to Existing Methods

Table 2 (and 4 in section B.3) presents the results for single-class forgetting across CIFAR-10 using ResNet18 (and VGG19). In addition to TRW, we introduce TRW-2R, a lightweight variant that applies the same TRW loss but restricts gradient updates to only two randomly selected layers in the network, significantly reducing computational cost while retaining strong unlearning performance. Similar results for MNIST and CIFAR-100 are shown in section B.4 and evaluations on Tiny-ImageNet dataset for ResNet18 can be found in section B.5. Our results show that our method consistently outperforms other methods across all datasets and architectures. Importantly, the retained class accuracy ($ACC_r$) remains competitive with or superior to baseline unlearning methods, indicating minimal interference with the retained knowledge. Additionally, the MIA scores are either comparable to or better than prior work, suggesting that our method does not compromise membership privacy. These results demonstrate that while our approach is resilient to MIA-NN, it remains competitive to SOTA unlearning methods in common evaluation metrics used in prior work.

### 5.2 Stronger MIA evaluation

We evaluate our method under a stronger MIA using the U-LiRA framework (Hayes et al., 2024), which assumes an adversary with access to shadow models to assess whether unlearned models retain residual information about the forgotten class.

Following the U-LiRA protocol, we train three shadow ResNet18 models on CIFAR-10. Each is unlearned with consistent hyperparameters to generate shadow unlearned models. Separately, we retrain three additional models with the forget class excluded to serve as shadow Retrain models. The attacker is trained to distinguish whether a given prediction comes from an unlearned or Retrain model based on class-conditional statistics, and the learned decision boundary is applied in a leave-one-out manner. A perfect unlearning

| Data | Method | ResNet18 (He et al., 2016) | | | | |
|---|---|---|---|---|---|---|
| | | $ACC_r$ ($\uparrow$) | $ACC_f$ ($\downarrow$) | $MIA$ ($\uparrow$) | $MIA$-NN ($\uparrow$) | Avg gap |
| CIFAR-10 | Original | $94.74 \pm 0.09$ | $94.42 \pm 5.45$ | $0.02 \pm 0.02$ | – | – |
| | Retrain | $94.83 \pm 0.13$ | $0$ | $100 \pm 0$ | $95.27 \pm 10.92$ | $0.00$ |
| | FT (Warnecke et al., 2021) | $85.60 \pm 2.35$ | $0$ | $96.53 \pm 1.16$ | $84.78 \pm 11.57$ | $-5.80$ |
| | RL (Golatkar et al., 2020) | $84.74 \pm 4.25$ | $0$ | $94.99 \pm 1.82$ | $49.21 \pm 14.03$ | $-15.29$ |
| | GA (Thudi et al., 2022) | $90.25 \pm 0.28$ | $14.12 \pm 2.17$ | $96.70 \pm 0.10$ | $28.15 \pm 12.92$ | $-15.22$ |
| | l1 (Jia et al., 2023) | $93.21 \pm 0.13$ | $0.9 \pm 0.05$ | $100 \pm 0$ | $14.19 \pm 8.22$ | $-20.45$ |
| | BU (Chen et al., 2023) | $87.68 \pm 2.23$ | $0$ | $85.91 \pm 3.97$ | $25.73 \pm 6.17$ | $-22.70$ |
| | SalUn (Foster et al., 2024) | $92.11 \pm 0.65$ | $0$ | $96.33 \pm 2.37$ | $9.62 \pm 4.78$ | $-23.01$ |
| | SVD (Kodge et al., 2024) | $94.17 \pm 0.57$ | $0$ | $97.20 \pm 3.77$ | $67.19 \pm 17.79$ | $-7.89$ |
| | SCRUB (Kurmanji et al., 2023) | $91.07 \pm 0.79$ | $0$ | $85.01 \pm 1.02$ | $10.24 \pm 5.41$ | $-25.95$ |
| | SCAR (Bonato et al., 2024) | $93.57 \pm 0.03$ | $0$ | $95.87 \pm 3.58$ | $71.93 \pm 15.85$ | $-7.18$ |
| | l2ul (Cha et al., 2024) | $87.86 \pm 1.79$ | $0$ | $94.62 \pm 0.15$ | $32.18 \pm 8.17$ | $-18.86$ |
| | **TRW** | $94.28 \pm 0.47$ | $0$ | $97.65 \pm 2.06$ | $95.82 \pm 13.72$ | $-0.59$ |
| | **TRW-2R** | $94.39 \pm 2.05$ | $0.32 \pm 0.32$ | $96.35 \pm 0.65$ | $94.67 \pm 12.47$ | $-1.09$ |

Table 2: Results on CIFAR-10 for ResNet18. Avg gap is the average difference across $(ACC_r, ACC_f, MIA, MIA\text{-NN})$ relative to the retrain baseline.

method should yield 50% accuracy—indicating the accuracy of a random classifier. Based on the results reported in Table 3, our methods achieve the lowest U-LiRA accuracy, demonstrating strong resilience to adaptive attacks.

| Metric | TRW | TRW-2R | FT | RL | GA | SalUn | SVD | l1 | BU | SCRUB | SCAR | l2ul |
|---|---|---|---|---|---|---|---|---|---|---|---|---|
| U-LiRA Accuracy (%) | 71.12 | **67.72** | 72.57 | 96.79 | 84.47 | 97.50 | 79.30 | 98.32 | 81.08 | 71.91 | 92.38 | 85.67 |

Table 3: U-LiRA Membership Inference Attack accuracy on CIFAR10 with ResNet18. Lower is better (50% indicates ideal unlearning). TRW-2R and TRW achieve the best performance.

# 6 LIMITATIONS & FUTURE WORK

Although we demonstrate the effectiveness of our approach through various empirical studies, the evidence presented is entirely empirical. Further theoretical work could provide deeper insights into the effectiveness of the tilted reweighting objective and the design of appropriate scoring functions to better approximate the distribution of retrained models. The scoring function used in this work is sample-independent; evaluating finer-grained scores that account for individual samples could be an interesting direction for future research. Moreover, adding higher-order constraints could be studied for capturing the systematic bias introduced in the retrained models. While MIA-NN reveals shortcomings in prior work, it remains heuristic. But we believe it can serve as a great complement to other membership attacks and evaluation metrics for future work in class unlearning. Further improvement in this method could be another interesting avenue for exploration. Finally, we should note that the underlying ideas in TRW can be generalized to other models, such as diffusion models and GANs (see C for details).

# 7 CONCLUSION

We introduce *Tilted ReWeighting* objective for class unlearning, a lightweight technique that erases an entire class from a pretrained classifier by redistributing the forgotten class's probability mass while accounting for the systematic bias of the retrained models towards more similar classes. To quantify how closely an unlearned model is to a fully retrained baseline, we propose *membership-inference attack via nearest neighbor* (**MIA-NN**) that utilizes residual mis-mapping of forgotten outputs: previous unlearning methods that pass standard tests fail under this stronger evaluation, whereas our tilted reweighting approach (TRW) remains robust. Experiments on CIFAR-10/100 and Tiny-ImageNet demonstrate that a few epochs of fine-tuning using TRW achieves retraining-level performance and is robust to existing MIAs and MIA-NN.

## 8 Reproducibility statement

All algorithmic details of the proposed MIA-NN attack and Tilted ReWeighting (TRW) method are described in Section 3, and the theoretical results are accompanied by detailed proofs in section A.2. We provide complete information about datasets, data preprocessing steps, and model architectures in Section section 4.1, while comprehensive descriptions of baseline methods are included in Appendix section B.1. Our experimental setup, including training schedules, optimization parameters, and evaluation metrics, is fully documented in section B.2. Additionally, we release an anonymized implementation of our code and scripts used to reproduce all results reported in the paper as supplementary material. Even the implementation for regenerating the results of the toy example presented in Figure 2 is included in our submitted code.

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

APPENDIX

# A  METHODS (CONT.)

In Section A.1 we present the formalism for the underlying threat model used in the design of MIA-NN. In Section A.2 we present the proof of Proposition 3.1 and in Section A.3 we will see further empirical observation about the necessity of using the tilted distribution to better approximate the distirubution of the Retrain model. In Section A.4 we will elaborate on the scoring function used in our experiments.

## A.1  THREAT MODEL

MIA-NN extends the setting of U-MIA for evaluation of unlearning a random set of points given in [1] to the setting of class unlearning. MIA on class unlearning methods can be considered as the following game that instantiates an adversary:

1. Assume a dataset $\mathcal{D}$ containing a set of labels $\mathcal{Y}$, a forget class $y_f$ (with corresponding samples $\mathcal{D}_f \in \mathcal{D}$) and model $\mathcal{M}$.
2. The challenger trains a model on $\mathcal{D}$ to derive parameters $\theta_0$. It then applies an unlearning algorithm to the model to unlearn class $y_f$ and derive parameters $\theta_u$. The challenger also trained a model on $\mathcal{D} - \mathcal{D}_f$ to derive parameters $\theta_t$.
3. Next, the challenger flips a fair coin $b \in \{0, 1\}$, and send to the adversary $(\theta_t, y_f)$ if $b = 0$ and $(\theta_u, y_f)$ if $b = 1$.
4. The adversary creates a decision rule $h : (\theta, y) \to \{0, 1\}$ that predicts whether the training data for deriving $\theta$ had included the class $y$ in it. The adversary wins if $h(\theta, y) = b$.

Basically, in this attack, we assume that we have an unlearned model and an adversary who wants to infer whether this model has been trained without the target class or is a result of an ineffective unlearning method; this adversary could be also the user who has requested the unlearning and wants to verify that their request has been performed correctly by the model owner. We assume only the query access to model outputs (probabilities) and some public test data from all classes in $\mathcal{Y}$. Note that, for the last part of the threat model, the adversary compares the computed MIA-NN accuracy on the unlearned model to the the the values derived from a set of shadow retrained models, and decides how much the MIA-NN accuracy from the unlearned model is similar to the retrained ones.

Many prior MIAs adapted to the setting of unlearning for evaluations assume that the adversary has also access to the training data ($\mathcal{D} - \mathcal{D}_f$) to derive logits values of the model on the training samples or train shadow models for the retrained model Fan et al. (2023); Chen et al. (2023). However, our proposed attack also works in the setting of black-box attack where we drop this assumption and utilize the shadow retrained models trained by the adversary on public data (disjoint from the training set). For further details on this setting see section B.9.

## A.2  PROOFS

*Proof of Proposition 3.1.* We start with showing that it is enough to prove the proposition holds for $\tilde{p}$ (i.e., $q^*$ is the I-projection of $\tilde{p}$ onto the probability simplex of retrained classes with the given linear constraints).

Let $q \in \mathcal{A}$; then $q(f \mid x) = 0$ and $q$ has support only on retained labels. Note that:

$$\mathrm{KL}(q \,\|\, p) = \sum_{y \neq f} q(y) \log \frac{q(y)}{p(y)} = \sum_{y \neq f} q(y) \log \frac{q(y)}{\tilde{p}(y)(1 - p(f))} = \sum_{y \neq f} q(y) \log \frac{q(y)}{\tilde{p}(y)} - \log\big(1 - p(f)\big).$$

Now, the final term is independent of $q$, hence

$$\arg\min_{q \in \mathcal{A}} \mathrm{KL}(q \,\|\, p) = \arg\min_{q \in \mathcal{A}} \mathrm{KL}(q \,\|\, \tilde{p}).$$

Therefore, the information projection from $p$ onto the set $\mathcal{A}$ is equivalent to the projection from $\tilde{p}$ onto this set. Now, the objective $\mathrm{KL}(q\|\tilde{p}) = \sum_y q(y)\log\big(q(y)/\tilde{p}(y)\big)$ is strictly convex on the simplex, so any feasible minimizer is unique. Now we introduce Lagrange multipliers $\alpha, \beta$ for the normalization and expectation constraints. Stationarity with respect to $q(y)$ gives

$$\log \frac{q(y)}{\tilde{p}(y)} + 1 + \alpha + \beta s_y = 0,$$

which rearranges to

$$q(y) = \tilde{p}(y)\,\exp\{-1 - \alpha - \beta s_y\}.$$

Normalizing yields the exponential family

$$q_\beta(y) = \frac{\tilde{p}(y)\,e^{\beta s_y}}{\sum_j \tilde{p}(j)\,e^{\beta s_j}}.$$

It remains to pick $\beta$ so that $\sum_y q_\beta(y)\,s_y = c$. First, we define $m(\beta) = \sum_y q_\beta(y) s_y$. So we need to find value of $\beta$ that gives us $m(\beta) = c$. We suppress the fixed $x$ to lighten notation and define:

$$Z(\beta) := \sum_{j \in S} \tilde{p}(j)\,e^{\beta s_j}, \qquad q_\beta(y) := \frac{\tilde{p}(y)\,e^{\beta s_y}}{Z(\beta)},$$

Now note that:

$$Z'(\beta) = \sum_{j \in S} \tilde{p}(j)\,s_j\,e^{\beta s_j}, \qquad Z''(\beta) = \sum_{j \in S} \tilde{p}(j)\,s_j^2\,e^{\beta s_j}.$$

So, we can write:

$$m(\beta) = \sum_{y \in S} \frac{\tilde{p}(y)\,e^{\beta s_y}}{Z(\beta)}\,s_y = \frac{Z'(\beta)}{Z(\beta)}.$$

Therefore, since $Z(\beta)$ is $C^\infty$, $m(\beta)$ is $C^\infty$ as well, and by differentiating once more we get:

$$m'(\beta) = \frac{Z''(\beta)}{Z(\beta)} - \left(\frac{Z'(\beta)}{Z(\beta)}\right)^2 = \sum_{y \in S} q_\beta(y)\,s_y^2 - \left(\sum_{y \in S} q_\beta(y)\,s_y\right)^2 = \mathrm{Var}_{q_\beta}(s) \geq 0.$$

Hence $m$ is nondecreasing on $\mathbb{R}$. Moreover, if the scores are not all equal on $S$, then $q_\beta(y) > 0$ for all $y \in S$, and the variance $\mathrm{Var}_{q_\beta}(s)$ is strictly positive, so

$$\forall \beta \in \mathbb{R}: \qquad m'(\beta) = \mathrm{Var}_{q_\beta}(s) > 0,$$

i.e., $m$ is *strictly* increasing. Moreover, $\lim_{\beta \to -\infty} m(\beta) = \min s_y$ and $\lim_{\beta \to +\infty} m(\beta) = \max s_y$. Thus, by the intermediate value theorem, for any feasible $c$ there exists a unique $\beta^\star$ with $m(\beta^\star) = c$. The corresponding $q^\star = q_{\beta^\star}$ is the unique minimizer. $\qquad\square$

### A.3 Effect of tilting

In this section, we further evaluate the insufficiency of the basic rescaling of the target model's probability distribution to approximating the distribution of the Retrain model. For this purpose, we plot the conditional distributions of each remaining classes, when unlearning the `automobile` (Figure 3) and `frog` (Figure 4) classes from a ResNet18 model trained on CIFAR-10. As shown in both figures, the rescaled conditional distributions are very different from those of the Retrain model. As mentioned earlier, in the Retrain model, the distributions are more skewed toward a few classes. This bias toward more similar classes are better captured using the tilted reweighting distribution, which utilizes the inter-class similarities to adjust the rescaled distribution by introducing the bias toward more similar classes that is expected from the Retrain model.

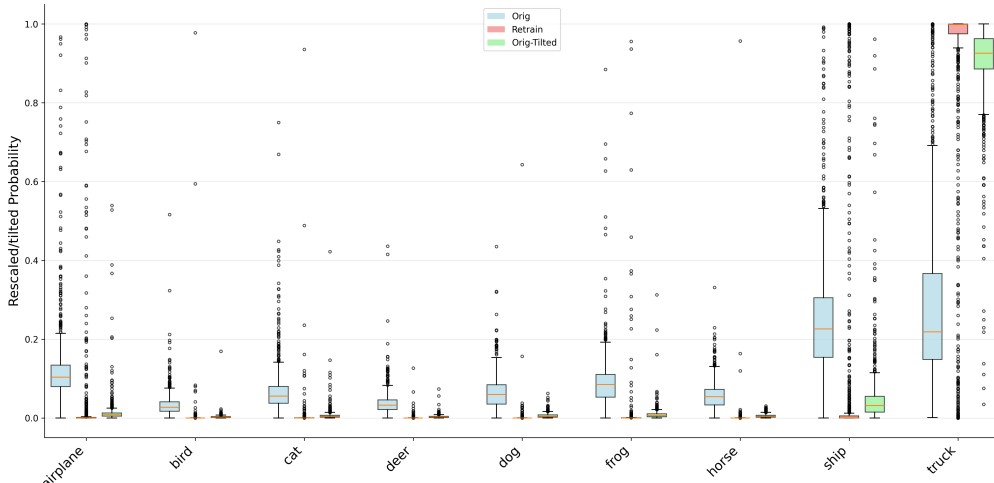

Figure 3: Forgetting the `automobile` class in a ResNet-18 model trained on CIFAR-10. The reweighted probabilities of the original model according to equation 1 (`Orig`) compared to the probabilities assigned to `automobile` in the Retrain model. As the figure shows, the Retrain model has a much higher bias toward `Truck` class, which is better captured when using the tilted reweighting accoridng to equation 2 (`Orig-Tilted`).

### A.4 Score function

Our goal is to assign, for each retained class $y \neq y_f$, a scalar similarity score $s_y$ that captures how likely a classifier trained *without* the forget class $y_f$ would bias predictions of $x \in D_f$ toward class $y$. We use a geometry-based score derived from the classifier's logit weights. Let $w_y \in \mathbb{R}^d$ denote the column of the final linear layer (logit weights) for class $y \in \mathcal{Y}$ in the original model. To capture the dominant inter-class structure while reducing noise and redundancy, we perform PCA on the matrix $W = [\, w_1 \; \cdots \; w_K \,] \in \mathbb{R}^{d \times K}$. Let $U_{d'} \in \mathbb{R}^{d \times d'}$ be the top $d'$ principal directions ($d' \ll d$) and define the projected class embeddings

$$\phi(w_y) \triangleq U_{d'}^\top w_y \in \mathbb{R}^{d'}.$$

We then define cosine similarities between the forget class and each retained class:

$$\tilde{s}_y \triangleq \cos\big(\phi(w_y),\, \phi(w_{y_f})\big) = \frac{\langle \phi(w_y),\, \phi(w_{y_f}) \rangle}{\|\phi(w_y)\|_2 \, \|\phi(w_{y_f})\|_2}, \quad y \neq y_f. \tag{5}$$

Finally, we use softmax to derive $s_y$ values in the form of probabilities from the values $\tilde{s}_y$. We use a small value for the temperature of softmax in our experiments (e.g., $0.01$) to make the similarity values distinct for more similar classes.

## B  Experiments (cont.)

### B.1  Baseline Methods

In this section, we detail several baseline methods for machine unlearning.

**Retraining** refers to training a new model from scratch using only the retained dataset $\mathcal{D}_r$. This baseline serves as the ideal unlearning outcome and is commonly regarded as the "gold standard."

**Fine-tuning (FT)** (Warnecke et al., 2021) fine-tunes the source model on the remaining dataset $\mathcal{D}_r$. In contrast to this standard fine-tuning, our approach performs targeted intervention by adjusting the model's output distribution to explicitly suppress the forgotten class and reallocate its probability mass proportionally to non-forgotten classes.

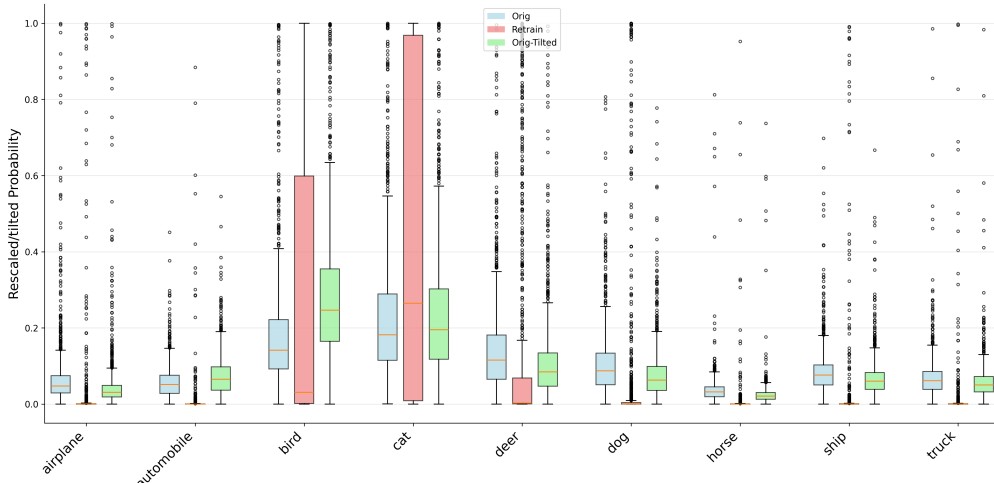

Figure 4: Forgetting the `frog` class in a ResNet-18 model trained on CIFAR-10. The reweighted probabilities of the original model according to equation 1 (`Orig`) compared to the probabilities assigned to `frog` in the Retrain model. As the figure shows, the Retrain model has a much higher bias toward a few classes, which are better captured when using the tilted reweighting accoridng to equation 2 (`Orig-Tilted`).

**Random Labeling (RL)** (Golatkar et al., 2020) fine-tunes the model with randomly relabeled forgetting data $\mathcal{D}_f$, which helps prevent the forgotten data from influencing the model's predictions.

**Gradient Ascent (GA)** (Thudi et al., 2022) inverts SGD updates to erase the influence of specific training examples.

**l1-sparse (l1)** (Jia et al., 2023) proposes a sparsity-aware unlearning framework that prunes model weights to simplify the parameter space.

**Boundary Unlearning (BU)** (Chen et al., 2023) shifts the decision boundary of the original model to mimic the model's decisions after retraining from scratch.

**SALUN (Fan et al., 2023)** introduce the concept of weight saliency and perform unlearning by modifying the model weights rather than the entire model, improving effectiveness and efficiency.

**SVD Unlearning (SVD)** (Kodge et al., 2024) performs class unlearning by identifying and suppressing class-discriminative features through singular value decomposition (SVD) of layer-wise activations.

**SCRUB** (Kurmanji et al., 2023) proposes a novel teacher-student formulation, where the student model selectively inherit from a knowing-all teacher only when the knowledge does not pertain to the data to be deleted.

**L2UL** (Cha et al., 2024) introduces an instance-wise unlearning framework that removes information using only the pre-trained model and the data points flagged for deletion.

**SCAR** (Bonato et al., 2024) proposes a distillation-trick mechanism that transfers the original model's knowledge to the unlearned model using out-of-distribution images, preserving test performance without any retain set.

## B.2 Experiment Settings

We evaluate our method using two standard architectures: **VGG-19** (Simonyan & Zisserman, 2014) and **ResNet-18** (He et al., 2016). All models are trained from scratch for **101 - 201 epochs** with a batch size of **128**. Optimization is performed using **stochastic gradient descent (SGD)**

with a learning rate of **0.1**, **momentum of 0.9**, and **weight decay of** $5 \times 10^{-4}$. The learning rate follows a `torch.optim.lr_scheduler.StepLR` schedule with `step_size` $= 40$ and `gamma` $= 0.1$.

For our unlearning procedure, we run updates for **10 epochs** with a learning rate of **0.001**. Baseline re-training budgets mirror prior work: GA (Thudi et al., 2022), FT (Warnecke et al., 2021), l1 (Jia et al., 2023), and L2UL (Cha et al., 2024) baselines are run for **20 epochs**, SVD (Kodge et al., 2024), and SCRUB (Kurmanji et al., 2023) are run for **10 epochs**, while SalUn (Fan et al., 2023) is run for **15 epochs** and SCAR (Bonato et al., 2024) for **25 epochs**. All experiments are implemented in **PyTorch 3.11** and executed on four **NVIDIA A40** GPUs, and repeated with three different random seeds; we report the average results across those runs.

### B.3 MORE RESULTS ON CIFAR-10

In section 5.1 we presented the results for ResNet-18 models trained on CIFAR-10. In Table 4 we present similar results for VGG19 models. As the table shows, our proposed methods outperform existing unlearning methods.

| Data | Method | VGG19 (Simonyan & Zisserman, 2014) | | | | |
|------|--------|----------------|----------------|----------------|----------------|---------|
| | | $ACC_r$ (↑) | $ACC_f$ (↓) | $MIA$ (↑) | $MIA$-NN (↑) | Avg gap |
| CIFAR-10 | Original | $92.68 \pm 0.05$ | $92.14 \pm 6.80$ | $0$ | $-$ | $-$ |
| | Retrain | $93.45 \pm 0.10$ | $0$ | $100 \pm 0$ | $94.10 \pm 9.66$ | $0.00$ |
| | FT (Warnecke et al., 2021) | $89.33 \pm 1.86$ | $0$ | $96.94 \pm 0.91$ | $79.69 \pm 12.81$ | $-5.40$ |
| | RL (Golatkar et al., 2020) | $85.57 \pm 1.29$ | $0$ | $94.07 \pm 0.90$ | $52.11 \pm 17.14$ | $-13.95$ |
| | GA (Thudi et al., 2022) | $87.26 \pm 0.30$ | $14.4 \pm 1.59$ | $97.10 \pm 0.87$ | $27.36 \pm 10.65$ | $-15.36$ |
| | l1 (Jia et al., 2023) | $90.12 \pm 2.18$ | $0.3 \pm 0.03$ | $92.61 \pm 7.23$ | $11.44 \pm 6.39$ | $-23.27$ |
| | BU (Chen et al., 2023) | $87.32 \pm 3.08$ | $0$ | $85.94 \pm 3.71$ | $18.92 \pm 4.03$ | $-23.84$ |
| | SalUn (Foster et al., 2024) | $89.76 \pm 1.00$ | $0$ | $97.35 \pm 0.65$ | $8.59 \pm 4.01$ | $-22.96$ |
| | SVD (Kodge et al., 2024) | $91.34 \pm 0.45$ | $0$ | $98.10 \pm 1.90$ | $65.42 \pm 11.54$ | $-8.17$ |
| | SCRUB (Kurmanji et al., 2023) | $89.84 \pm 1.16$ | $0$ | $84.28 \pm 1.91$ | $14.07 \pm 6.93$ | $-24.84$ |
| | SCAR (Bonato et al., 2024) | $92.59 \pm 1.80$ | $0$ | $98.06 \pm 2.11$ | $69.51 \pm 13.45$ | $-6.85$ |
| | l2ul (Cha et al., 2024) | $89.15 \pm 0.75$ | $0$ | $96.82 \pm 1.45$ | $35.90 \pm 9.46$ | $-16.42$ |
| | **TRW** | $91.58 \pm 0.23$ | $0$ | $99.26 \pm 0.74$ | $94.63 \pm 13.34$ | $-0.52$ |
| | **TRW-2R** | $91.91 \pm 0.63$ | $0$ | $99.52 \pm 0.74$ | $93.58 \pm 14.82$ | $-0.63$ |

Table 4: Results on CIFAR-10 for VGG19. Avg gap is the average difference across $(ACC_r, ACC_f, MIA, MIA\text{-NN})$ relative to the retrain baseline.

To further evaluate the effectiveness of our method in making the original model more similar to the retrain models, we conducted a study to evaluate the high-level geometry of the decision boundaries by looking at the class-wise clustering of samples embedding. Figure 5 shows the t-SNE plot of the CIFAR-10 embeddings derived from a trained ResNet-18 model before and after unlearning as a comparison of how these embeddings look like in retrain models. As the figures show, in the original model, the classes are well-separated in the embedding space into distinct clusters. In the retrain model, where class 1 (automobile) has been removed from the training data, the clusters corresponding to class 1 (automobile) and class 10 (Truck) have been merged. The third figure, which shows that our method TRW effectively replicates this behavior.

### B.4 RESULTS ON MNIST AND CIFAR-100

To study performance on a higher-variance image domain, we further evaluate on **MNIST** and **CIFAR-100** with VGG and ResNet-18 backbones. As shown in table 5, our methods deliver (near-)perfect forgetting with competitive retained accuracy and consistently stronger resistance to MIA attacks than the baselines.

### B.5 RESULTS ON TINY-IMAGENET-200

To evaluate unlearning on a more challenging benchmark, we also apply our method to the **Tiny-ImageNet-200** dataset using a ResNet-18 backbone. We average the results

| Data | Method | VGG19 (Simonyan & Zisserman, 2014) | | | ResNet18 (He et al., 2016) | | |
|---|---|---|---|---|---|---|---|
| | | $ACC_r$ (↑) | $ACC_f$ (↓) | $MIA$ (↑) | $ACC_r$ (↑) | $ACC_f$ (↓) | $MIA$ (↑) |
| MNIST | Original | 99.52 ± 0.01 | 99.57 ± 1.33 | 0 | 99.65 ± 0.04 | 99.91 ± 1.05 | 0.23 ± 0.23 |
| | Retraining | 99.54 ± 0.03 | 0 | 100 ± 0 | 99.64 ± 0.05 | 0 | 100 ± 0 |
| | FT (Warnecke et al., 2021) | 99.43 ± 0.72 | 0 | 99.25 ± 0.11 | 98.16 ± 0.65 | 0 | 95.78 ± 1.28 |
| | RL (Golatkar et al., 2020) | 99.04 ± 0.19 | 0 | 99.66 ± 0.21 | 99.33 ± 0.25 | 0 | 99.64 ± 0.15 |
| | GA (Thudi et al., 2022) | 97.69 ± 2.42 | 0 | 96.86 ± 2.87 | 98.26 ± 0.12 | 14.94 ± 0.03 | 85.19 ± 0.17 |
| | l1 (Jia et al., 2023) | 94.07 ± 4.48 | 0.01 ± 0.02 | 92.55 ± 1.53 | 93.47 ± 1.98 | 0.04 ± 0.02 | 97.51 ± 0.42 |
| | BU (Chen et al., 2023) | 93.14 ± 8.19 | 0 | 95.40 ± 0.06 | 94.12 ± 6.51 | 0 | 98.62 ± 0.15 |
| | SalUn (Foster et al., 2024) | 99.23 ± 0.18 | 0 | 100 ± 0 | 99.43 ± 0.77 | 0 | 100 ± 0 |
| | SVD (Kodge et al., 2024) | 99.16 ± 0.20 | 0 | 100 ± 0 | 99.37 ± 0.32 | 0 | 99.87 ± 0.13 |
| | SCRUB (Kurmanji et al., 2023) | 99.34 ± 0.03 | 0 | 98.73 ± 0.15 | 99.45 ± 0.06 | 0 | 91.88 ± 0.34 |
| | SCAR (Bonato et al., 2024) | 98.93 ± 0.10 | 0 | 100 ± 0 | 99.20 ± 0.24 | 0 | 97.82 ± 0.68 |
| | l2ul (Cha et al., 2024) | 99.15 ± 0.12 | 0 | 100 ± 0 | 95.75 ± 0.16 | 0 | 93.83 ± 0.19 |
| | **TRW** | **99.52 ± 0.07** | **0** | **100 ± 0** | 99.49 ± 0.04 | 0 | 100 ± 0 |
| | TRW-2R | 99.48 ± 0.13 | 0 | 99.78 ± 0.22 | 99.45 ± 0.08 | 0.04 ± 0.06 | 99.98 ± 0.15 |
| CIFAR-100 | Original | 69.87 ± 0.80 | 70.72 ± 6.41 | 0.45 ± 0.85 | 78.52 ± 0.58 | 78.93 ± 5.77 | 0.3 ± 0.5 |
| | Retraining | 69.54 ± 0.92 | 0 | 100 ± 0 | 78.30 ± 0.84 | 0 | 100 ± 0 |
| | FT (Warnecke et al., 2021) | 65.26 ± 1.58 | 0 | 87.41 ± 3.32 | 73.37 ± 1.42 | 0 | 86.09 ± 0.46 |
| | RL (Golatkar et al., 2020) | 58.83 ± 3.04 | 0 | 87.20 ± 3.11 | 70.97 ± 1.68 | 0 | 88.65 ± 0.33 |
| | GA (Thudi et al., 2022) | 62.34 ± 0.77 | 0 | 83.58 ± 0.41 | 72.05 ± 0.19 | 0 | 85.11 ± 1.45 |
| | l1 (Jia et al., 2023) | 57.28 ± 1.47 | 0 | 83.89 ± 1.78 | 72.30 ± 1.84 | 0 | 84.75 ± 2.06 |
| | BU (Chen et al., 2023) | 59.27 ± 3.23 | 0 | 84.32 ± 2.45 | 67.52 ± 1.86 | 0 | 81.62 ± 0.79 |
| | SalUn (Foster et al., 2024) | 63.58 ± 3.06 | 0 | 99.97 ± 0.03 | 72.11 ± 1.37 | 0 | 98.65 ± 1.35 |
| | SVD (Kodge et al., 2024) | 68.53 ± 1.78 | 0 | 100 ± 0 | 75.86 ± 1.79 | 0 | 99.30 ± 0.08 |
| | SCRUB (Kurmanji et al., 2023) | 58.91 ± 1.32 | 0 | 89.01 ± 1.34 | 76.92 ± 1.06 | 0 | 87.65 ± 2.18 |
| | SCAR (Bonato et al., 2024) | 65.97 ± 1.88 | 1.38 ± 1.04 | 100 ± 0 | 75.42 ± 1.17 | 0.5 ± 0.43 | 97.61 ± 0.81 |
| | l2ul (Cha et al., 2024) | 66.77 ± 0.84 | 0 | 99.52 ± 0.53 | 71.56 ± 1.38 | 0 | 97.05 ± 1.97 |
| | **TRW** | 69.36 ± 0.61 | 0 | 99.15 ± 0.75 | **78.05 ± 0.43** | **0** | **99.56 ± 1.75** |
| | TRW-2R | 69.07 ± 1.69 | 0 | 99.00 ± 1.36 | 77.97 ± 0.78 | 0.01 ± 0.07 | 98.87 ± 1.15 |

Table 5: **Single-class forgetting on MNIST and CIFAR-100** We bold the method with the highest retained accuracy ($ACC_r$), membership attack robustness (MIA), and lowest forgotten class accuracy ($ACC_f$). Our method (TRW, TRW-2R) consistently achieves perfect forgetting ($ACC_f = 0$), while preserving high retained accuracy and strong MIA robustness across datasets and architectures.

| Method | ResNet-18 | | |
|---|---|---|---|
| | $ACC_r$ (↑) | $ACC_f$ (↓) | $MIA$ (↑) |
| Original | 63.49 | 64.52 | 0 |
| Retrain | 63.32 | 0 | 100 |
| FT (Warnecke et al., 2021) | 58.92 | 0 | 83.62 |
| RL (Golatkar et al., 2020) | 41.58 | 0 | 83.55 |
| SCRUB (Kurmanji et al., 2023) | 62.31 | 0 | 87.73 |
| SCAR (Bonato et al., 2024) | 61.48 | 0.63 | 97.29 |
| l2ul (Cha et al., 2024) | 61.42 | 0 | 96.18 |
| TRW | 62.78 | 0 | 100 |
| **TRW-2R** | **62.84** | **0** | **100** |

Table 6: **Single-class forgetting on Tiny-ImageNet-200 with ResNet-18.** We report retained accuracy ($ACC_r$), forget accuracy ($ACC_f$), and membership inference attack accuracy (MIA).

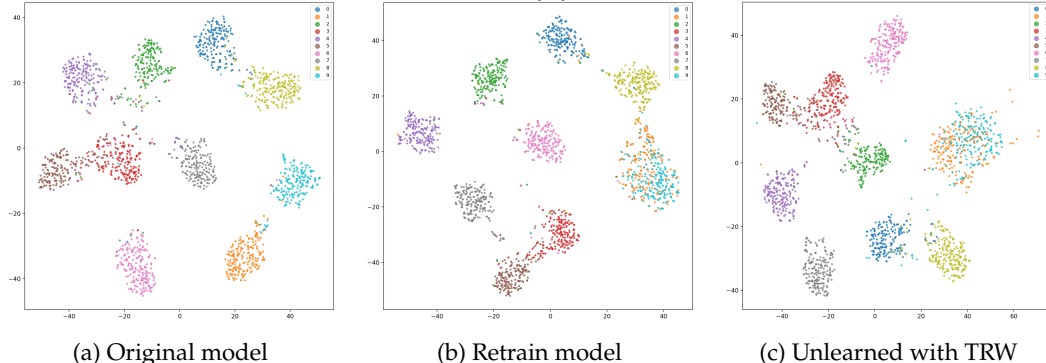

|                        |                        |                        |
|:----------------------:|:----------------------:|:----------------------:|
| (a) Original model     | (b) Retrain model      | (c) Unlearned with TRW |

Figure 5: The figures shows the embedding of test samples (unseen during training) of CIFAR-10 derived from a ResNet18 model for the (a) original model, (b) retrain model, and (c) the original model after applying TRW for unlearning. As the figure shows the retrain model that has been trained without `class 1` (automobile), mixes the embeddings of the samples from that class with `class 9` (truck). TRW effectively modifies the original model to replicate this behavior.

over 10 semantically diverse classes: *goldfish*, *European fire salamander*, *bullfrog*, *tailed frog*, *American alligator*, *boa constrictor*, *trilobite*, *scorpion*, *black widow spider*, and *tarantula*. Due to computational constraints, we compare against a selected subset of representative baselines. From table 6, we observe that our methods achieve perfect forgetting with strong retained accuracy and MIA robustness.

## B.6 RUNNING TIME ANALYSIS

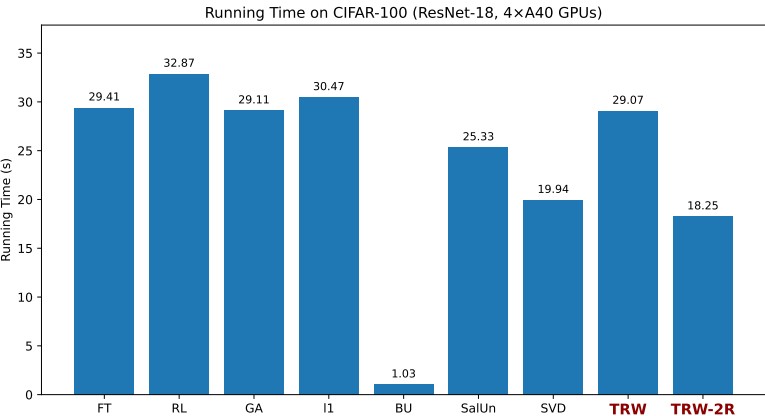

Figure 6: Running time comparison (in seconds per epoch) on CIFAR-100 with ResNet-18 using 4×A40 GPUs. Our TRW-2R and TRW are among the fastest methods.

We compare the wall-clock training time of each unlearning method on the CIFAR-100 dataset using a ResNet-18 backbone, measured on four NVIDIA A40 GPUs. As shown in fig. 6, our proposed **TRW-2R** method is among the fastest, completing in 18.25s, which is significantly faster than standard retraining-based baselines like FT and RL.

Importantly, both of our methods — TRW, and TRW-2R — achieve optimal unlearning performance within just **10 epochs**. This makes them highly efficient and scalable in practice, offering strong privacy guarantees at a fraction of the computational cost of full retraining.

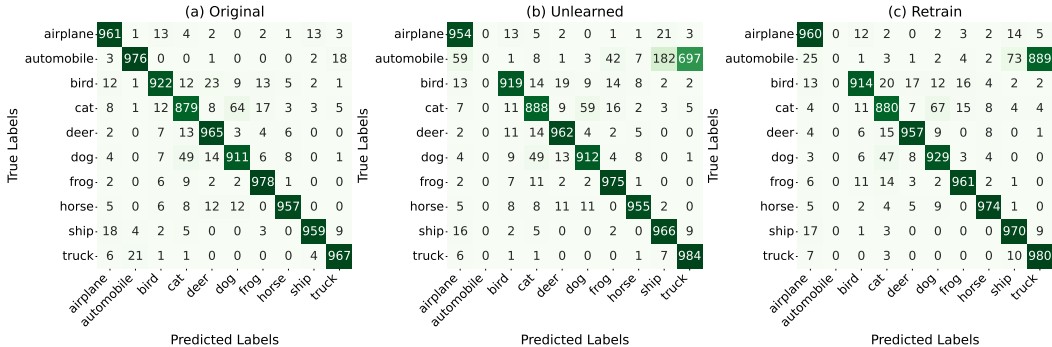

Figure 7: Confusion Matrix for original ResNet18 model, our unlearning model and retraining model forgetting automobile class, showing redistribution of automobile samples to other classes in proportion to the confusion in original model.

### B.7 CONFUSION MATRIX ANALYSIS

Following Kodge et al. (2024), we plot the confusion matrix showing the distribution of true labels and predicted labels for the original ResNet18 model, our unlearning model (on Automobile) and retraining model for CIFAR10 in fig. 7. Interestingly, we observe that in our unlearning model, a great number of the automobile samples are assigned to 'truck' class, with a small portion being assigned to the 'ship' class, indicating that they share some similar features. This result aligns with that in the retraining model. Furthermore, we see no performance drop in the remaining classes, indicating the robustness of our method.

**Forgetting Class Dependency:** In addition to evaluating forgetting efficacy through classification accuracy, we further examine where the predictions of forgotten samples are redirected. Figure 8 summarizes the top classes that *willow tree* samples were reassigned to after unlearning. Notably, a significant proportion of the reassigned predictions fall into semantically or visually similar categories, such as *palm tree* (25%), *forest* (17%), and *oak tree* (14%). This suggests that although the model effectively forgets the target class, it redistributes the predictions to classes with similar visual features.

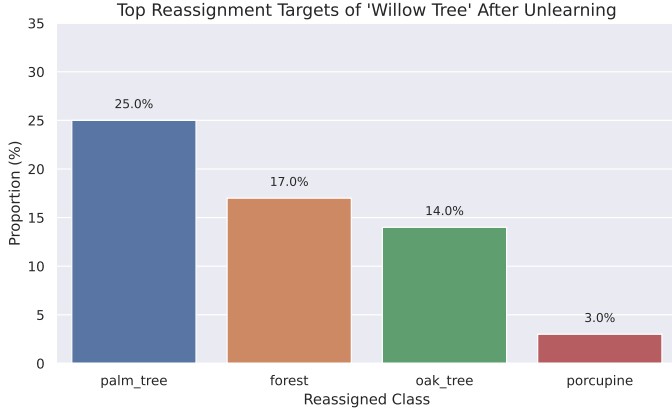

Figure 8: **Top reassignment targets of the forgotten class *willow tree*.** The model tends to reassign samples to semantically similar classes such as *palm tree*, *forest*, and *oak tree*.

In addition, section B.7 highlights the top five classes most affected by the unlearning of the *willow tree* class in CIFAR-100. Notably, *maple tree* suffers the largest performance degradation, with a 26% drop in accuracy. This is followed by moderate drops in *lion*, *otter*, *boy*, and *dinosaur*, which show decreases of 5–7%. These affected classes are likely to share

| Class Name | Original Accuracy (%) | Unlearned Accuracy (%) | Accuracy Drop (%) |
|---|---|---|---|
| maple tree | 73.00 | 47.00 | 26.00 |
| lion | 89.00 | 82.00 | 7.00 |
| otter | 60.00 | 53.00 | 7.00 |
| boy | 58.00 | 52.00 | 6.00 |
| dinosaur | 76.00 | 71.00 | 5.00 |

Table 7: **Top classes with the largest accuracy drop after unlearning the willow tree class in CIFAR100.**

visual similarities with *willow tree*, indicating that the unlearning process may have altered representations that were partially shared across conceptually related categories.

B.8 Ablation Study

| $\beta_f$ | $ACC_r$ ($\uparrow$) | $ACC_f$ ($\downarrow$) | $MIA$ ($\uparrow$) | $MIA - NN$($\uparrow$) |
|---|---|---|---|---|
| 0 | 94.35 | 0 | 99.94 | 47.2 |
| 5 | 94.37 | 0 | 98.94 | 76.1 |
| 10 | 94.28 | 0 | 97.65 | 82.1 |
| 20 | 77.67 | 0 | 98.72 | 98.6 |

Table 8: Effect of $\beta_f$ on retained accuracy ($ACC_r$), forgetting accuracy ($ACC_f$), and membership inference resistance ($MIA$ and $MIA\text{–}NN$). Increasing $\beta_f$ improves unlearning robustness but may harm retained performance. We select $\beta_f = 10$ as it provides the best balance between maintaining accuracy on retained data and achieving strong resistance to membership inference attacks.

We explore different choices of $\beta$ as it directly affects the trade-off between retained accuracy and unlearning robustness and the results are shown in Table 8. We find that $\beta = 10$ provides the most balanced result across metrics. At this setting, the retained accuracy ($ACC_r$) remains very high (94.28), almost identical to $\beta = 0$ and $\beta = 5$, indicating that the model preserves strong predictive performance on the retained data. Meanwhile, the unlearning effectiveness improves significantly: the $MIA$ score decreases to 97.65, and the $MIA\text{–}NN$ score rises to 82.1. This demonstrates that the model is more resistant to membership inference attacks compared to smaller $\beta$ values, where adversaries can more easily detect forgotten samples. Although $\beta = 20$ achieves even stronger robustness ($MIA\text{–}NN = 98.6$), it severely degrades retained accuracy (dropping to 77.67). Therefore, we select $\beta = 10$ as it achieves the optimal balance between maintaining accuracy on retained data and providing strong attack robustness.

In addition to the ablation study on the tilt parameter ($\beta$), we performed an ablation study on the other hyper-parameters of our method, including the choice of the class-wise similarity function, inverse temperature for deriving the class-wise similarity scores, and the number of dimensions in PCA. The results are presented in Table 9.

Our experiments showed that the use of cosine similarity of class-wise weight vectors in the last layer (82.1 MIA-NN) leads to empirically stronger and closer to the retrain ideal (90.1) compared to the Euclidean distances (77.8). It also shows that our model is robust to slight changes in PCA dimension, tilt parameter ($\beta$), and inverse temperature in computing the similarity scores. Although we fixed our hyper-parameters in all our experiments (different models, datasets, and forget classes) to: a) cosine similarity of the class-wise weight vectors, b) PCA dimension of 32, c) inverse temperature of 5, and d) $\beta = 10$, as the results show, hyper-parameter tuning can lead to slight improvements in some cases.

In addition to this regular ablation study, we conducted another experiment to show the effect of inverse temperature (for computing the similarity scores) and the tilt parameter ($\beta$) in our method when applied to unlearning the `automobile` class from a ResNet-18 model

| Hyperparameter | Value | $ACC_r$ | $ACC_f$ | MIA | MIA-NN |
|---|---|---|---|---|---|
| retrain | - | 94.83 | 0 | 100 | 90.1 |
| PCA dim | - | 93.75 | 0 | 99.12 | 86.8 |
| | 16 | 94.22 | 0 | 99.82 | 83.5 |
| | 32 | 94.28 | 0 | 97.65 | 82.1 |
| | 64 | 94.22 | 0 | 99.82 | 83.5 |
| Distance | Euclidean | 94.27 | 0 | 100 | 77.8 |
| | Cosine | 94.28 | 0 | 97.65 | 82.1 |
| $Inv_{temp}$ | 0.5 | 94.30 | 0 | 99.85 | 81.4 |
| | 1 | 94.32 | 0 | 99.12 | 81.0 |
| | 5 | 94.27 | 0 | 98.07 | 82.1 |
| | 10 | 91.85 | 0 | 97.88 | 86.1 |
| | 100 | 75.72 | 0 | 97.87 | 95.3 |

Table 9: Ablation study in forgetting the class `automobile` in CIFAR10. Unless otherwise noted, TRW uses cosine similarity of class-wise weight vectors as the class-wise similarity function and uses softmax with inverse temperature of 5 to derive the similarity scores. Default dimension for PCA is 32, and we use $\beta$=10.

trained on CIFAR-10. In this experiment, we are interested in finding out how the weight given to the nearest-neighbor class (`truck`) in equation 2 changes. Basically, we compute $\frac{\exp(\beta\,s_y)}{\sum_{j\neq y_f}\exp(\beta\,s_j)}$ for the `truck` class. Note that for the rescaled distribution given in equation 1 (which is equivalent to setting $\beta = 0$ in equation 2), this value is equal to $\frac{1}{9} \approx 0.11$. As Figure 9 shows, for the default parameters used in our experiments ($\beta = 10$ and an inverse temperature of 5), this leads to a modest boost in the weight of the most similar class (0.15). This modest boost is due to the fact that the cosine similarity values that we compute for the class-wise weight vectors lead to very small differences for some of the classes. For example, for the `automobile` class, the cosine similarities with the three most similar classes `truck`, `ship`, and `airplane` are 0.937, 0.922, and 0.919, accordingly. Therefore, as shown in the figure and the presented table, our method is robust to modest changes in the value of $\beta$ and the inverse temperature and only deteriorates at extreme values.

Also note that a benefit of using the initial softmax for computing the similarities is that despite the choice of the similarity metric and the range of values, it converts the similarity values to probabilities in range $[0, 1]$ and therefore prevents extreme changes from appearing when computing the weights in the tilted distribution of equation 2.

## B.9 BLACKBOX SETTING

Although in our reported results the retrain models used by the adversary have been trained on the training set, we can drop this assumption and utilize the models retrained by the adversary on public data (disjoint from the training set).

To perform this experiment and show-case the performance of MIA-NN in a black-box setting, we randomly split the CIFAR-10 training set into two parts. We use only the first part for training the original model. The adversary has access to the second part, which is disjoint from the training samples and can be considered public data in this setting, and uses that for training one retrained model. Then the adversary utilizes the CIFAR-10 test set (which is again considered public data) and the output probabilities from the original model to evaluate whether the original model has been trained on the forgotten class. Therefore, in this setting, we have shown the use-case of MIA-NN in a black-box setting. Table 10 shows the results of black-box MIA-NN for revealing the current shortcomings in existing unlearning methods. As the results show, MIA-NN in the black-box setting is also very effective.

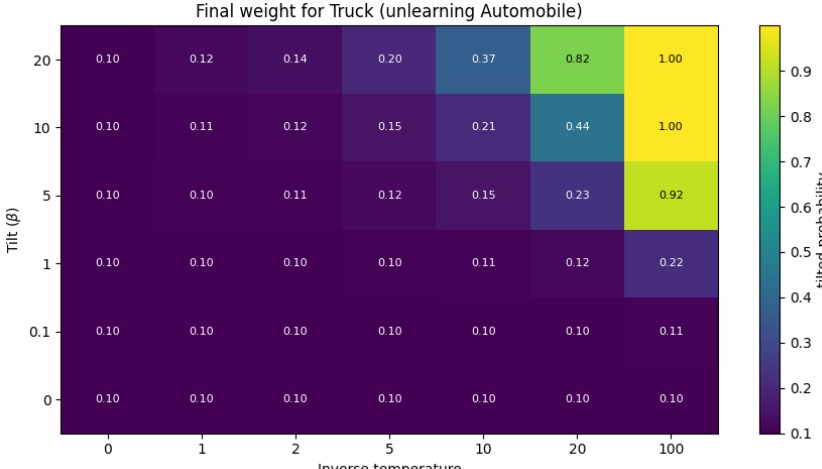

Figure 9: To see the actual effect of tilting the distribution based on the value of the inverse temperature and the tilt parameter ($\beta$), we plot the final weight applied to the conditional distribution $\tilde{p}(y \mid x)$ in equation 2 when $y$ is the `Truck` label. As the figure shows, for the default values of $\beta = 10$ and inverse temperature of 5 (fixed in our experiments), the weights are not very sharp and deviate slightly from the default 0.11 ($\frac{1}{9}$) weights assigned to each conditional probability in equation 1 (the rescaled probabilities). As shown in the figure and also the table above on the ablation studies, slight changes to the tilt parameter or inverse temperature does not make large differences.

| Dataset | Retrain | FT | RL | GA | SalUn | BU | l1 | SVD | SCRUB | SCAR | l2ul | TRW |
|---|---|---|---|---|---|---|---|---|---|---|---|---|
| CIFAR-10 (auto→ truck) | 83.1 | 66.8 | 26.1 | 12.8 | 2.11 | 11.6 | 9.2 | 47.9 | 9.4 | 55.2 | 21.5 | **86.1** |

Table 10: MIA-NN accuracy in the black-box setting. Lower gap with the Retrain models indicate more effective unlearning. The gap with the Retrain models reveals under-performance in many of the SOTA unlearning methods that have been evaluated using only regular MIAs.

If a public checkpoint for a model that has been trained on only the remaining classes is available to the adversary, the adversary would be able to utilize that model as it provides a similar setting to the one we used for our black-box setting (no access to the training data and the original model parameters). Still, this new black-box experiment, which requires only one retrained model on public data to reveal the shortcomings of existing unlearning methods is a rigorous and realistic threat model that also provides the community with a useful metric for evaluations of class unlearning methods.

### B.10 MULTIPLE CLASS FORGETTING

We evaluated our approach on multi-class forgetting tasks on CIFAR-100 using ResNet18, gradually increasing the number of randomly chosen forgotten classes from 1 to 10. Our method achieves stable and effective forgetting across different numbers of target classes. As shown in Table 11, the retained accuracy ($ACC_r$) remains consistent with the original model, while the forgotten class accuracy ($ACC_f$) drops to zero in all cases. Additionally, the MIA attack success rate remains near-perfect, demonstrating that the unlearned model is indistinguishable from retrained baselines even under strong adversarial probing. These results indicate the robustness of our forgetting strategy in multi-class scenarios.

| Forget Classes | ACC$_r$ ↑ | ACC$_f$ ↓ | MIA ↑ |
|---|---|---|---|
| 1 class | 77.08 (77.24) | 0 (77.03) | 100 (0.40) |
| 5 classes | 76.95 (77.10) | 0 (75.8) | 100 (0.96) |
| 10 classes | 74.84 (76.66) | 0 (77.04) | 99.42 (1.23) |

Table 11: **Multi-class forgetting performance on CIFAR-100 using ResNet18.** For each setting, we report the accuracy on the retained classes (ACC$_r$ ↑), the accuracy on the forgotten classes (ACC$_f$ ↓), and the MIA attack success rate (MIA ↑). Values in parentheses denote the corresponding metrics from the original (unforgotten) model. **Our method maintains high retained accuracy while fully forgetting target classes and preserving robustness under strong MIAs, demonstrating stable and scalable forgetting performance across varying numbers of classes.**

### B.11 The Use of Large Language Models (LLMs)

We used a large language model (LLM) strictly as a writing assistant for style and clarity. All LLM-suggested edits were reviewed line-by-line by the authors, and only adopted when they preserved the original meaning and technical correctness. And the LLM did not influence the scientific contributions, empirical results, or conclusions of this paper.

## C Future Work

Because of the differences in the nature of model training, objective, and definition of unlearning in these domains, LLMs, GNNs, and classification models have different approaches to unlearning. That being said, a few prior works in unlearning for classification models extend their approach to the setting of text-to-image generative models Fan et al. (2023). It is interesting to note that our method can be extended to the setting of text-to-image generative models. Consider a generative model (e.g., diffusion model) that has learned the conditional distribution $p_\theta(x|c_f)$ for the forget class/concept $c_f$. To unlearn from this model, we would be able to set the tilted distribution as follows:

$$q^\star(x|c_f) \propto \sum_{c \in \mathcal{C}_r} \exp(\beta s(c, c_f)) \, p_\theta(x|c)$$

As our target distribution, where $s(c, c_f)$ measures semantic similarity (e.g., CLIP embeddings) between the forget concept $c_f$ and the remaining concepts $c \in \mathcal{C}_r$ and $\beta$ controls the tilt. Using this tilted distribution as the target distribution would bias the model toward concepts that are more semantically similar to $c_f$, leading to utility preservation of the model while unlearning class $c_f$. In general, our framework is very flexible and various similarity measures could be used to derive the desired form of the target distribution.

