# OpenReview forum: "Silent Neighbors, Loud Secrets: Privacy Leakage from Nearby Classes in Unlearned Models"
_ICLR.cc/2026/Conference — Submitted to ICLR 2026_

### Official Review · Reviewer_HDYu · 2025-10-25

**Soundness:** 2
**Presentation:** 1
**Contribution:** 2
**Rating:** 4
**Confidence:** 5

**Summary:**

This paper argues that standard class-unlearning evaluations overlook class geometry, leaving leakage that a new attack (MIA-NN) can expose. The authors proposed TRW to mitigate the privacy leakage caused by MIA-NN and did extensive experiments to support their claims.

**Strengths:**

1. The nearest-neighbor leakage perspective is simple and intuitive.

2. TRW is an output-space objective, easy to implement, and adds little overhead relative to fine-tuning.

**Weaknesses:**

1. The MIA via nearest neighbors attack is not clearly substantiated in the main text. For an attack, at least the threat model, capability of each role and the attacking goal should be defined clearly. In this paper, it seems that all things are conducted by the ML server.


2. What MIA-NN really measures. As defined, MIA-NN trains per-class discriminators on retrain (and then on the unlearned model) and computes their accuracy on forget-class test samples to quantify a gap to retrain. This is not a per-example membership decision about whether a specific sample was in the original training set; it is a distributional test of whether forget-class behavior matches retrain. The paper sometimes presents MIA-NN as a “membership inference attack” without clearly distinguishing this from standard per-sample MIAs.


3. The proposed Tilted ReWeighting (TRW) objective adjusts the output distribution proportionally to class similarity, but this only ensures proportional alignment. It does not guarantee the decision boundary geometry or higher-order distributional structure. As a result, while the marginal behavior may resemble retraining, the actual local decision regions may diverge.

4. The paper's writing and methodological details are not clear, making it hard to follow:
 - In Section 3.4, the constant c (from the definition of set A) is not explained clearly (between Eq. 1 and Eq. 2, no marker). How to determine the value of expected similarity c, and why the paper set β as 10?
 - The theoretical motivation for using an exponential term (exp) in the tilting factor in Eq. 2, rather than a simpler linear weighting, is not discussed in the main text. The proof in the appendix shows this is a result of KL-divergence minimization, this should be mentioned after Eq.2 at least 1 sentence.
 - A method named TRW-2R appears in multiple tables (e.g., Table 2, 3) and often performs differently than TRW. However, this method is never defined or explained in the paper. This is a significant omission that needs to be addressed.

5. For paper citing, authors could use \citep{} to replace \cite{}.

**Questions:**

See weaknesses.

**Details Of Ethics Concerns:**

No ethics concerns

---

> ### Author Response · Authors · 2025-11-17
>
> We thank the reviewer for their insightful comments and feedback. We believe addressing these comments would improve the presentation quality of our work. Below are our responses to their questions and weaknesses:
>
> **W1. The threat model:** Thanks for pointing this out, we will polish the threat model in MIA-NN. MIA-NN extends the setting of U-MIA for evaluation of unlearning a random set of points given in [1] to the setting of class unlearning. MIA on class unlearning methods can be considered as the following game that instantiates an adversary:
>
> 1. Assume a dataset $\mathcal{D}$ containing a set of labels $\mathcal{Y}$, a forget class $y_f$ (with corresponding samples $\mathcal{D}_f \in \mathcal{D})$ and model $\mathcal{M}$.
>
> 2. The challenger trains a model on $\mathcal{D}$ to derive parameters $\theta_0$. It then applies an unlearning algorithm to the model to unlearn class $y_f$ and derive parameters $\theta_u$. The challenger also trained a model on $\mathcal{D} - \mathcal{D}_f$ to derive parameters $\theta_t$.
>
> 3. Next, the challenger flips a fair coin $b \in \{0,1\}$, and send to the adversary $(\theta_t, y_f)$ if $b=0$ and $(\theta_u, y_f)$ if $b=1$.
>
> 4. The adversary creates a decision rule $h: (\theta, y) \rightarrow \{0,1\}$ that predicts whether the training data for deriving $\theta$ had included the class $y$ in it. The adversary wins if $h(\theta,y) = b$.
>
>
> Basically, in this attack, we assume that we have an unlearned model and an adversary who wants to infer whether this model has been trained without the target class or is a result of an ineffective unlearning method; this adversary could be also the user who has requested the unlearning and wants to verify that their request has been performed correctly by the model owner. We assume only the query access to model outputs (probabilities) and some public test data from all classes in $\mathcal{Y}$. Note that, for the last part of the threat model, the adversary compares the computed MIA-NN accuracy on the unlearned model to the the values derived from a set of shadow retrained models, and decides how much the MIA-NN accuracy from the unlearned model is similar to the retrained ones.
>
> Many prior MIAs adapted to the setting of unlearning for evaluations assume that the adversary has also access to the training data ($\mathcal{D} - \mathcal{D}_f$) to derive logits values of the model on the training samples or train shadow models for the retrained model [2,3,4]. However, our proposed attack also works in the setting of black-box attack where we drop this assumption and utilize the shadow retrained models trained by the adversary on public data (disjoint from the training set).
>
> To show-case the performance of MIA-NN in a fully black-box setting, we randomly split the CIFAR-10 training set into two parts. We use only the first part for training the original model. The adversary has access to the remaining part, which is disjoint from the training samples and can be considered as public data in this setting, and uses that for training one retrained model. Then the adversary utilizes the CIFAR-10 test set (which again is considered public data) and the output probabilities from the original model to evaluate whether the original model has been trained on the forgotten class. Therefore, in this setting, we have shown the use-case of MIA-NN in a black-box setting. The following link shows the table of results for this setting, which is consistent with our prior results on revealing the shortcoming of existing methods.
>
> https://tinylink.net/v1fMv
>
> [1] Hayes, J., Shumailov, I., Triantafillou, E., Khalifa, A., \& Papernot, N. (2025, April). Inexact unlearning needs more careful evaluations to avoid a false sense of privacy. In 2025 IEEE Conference on Secure and Trustworthy Machine Learning (SaTML) (pp. 497-519). IEEE.
>
> [2] Fan, C., Liu, J., Zhang, Y., Wong, E., Wei, D., \& Liu, S. (2023, October). SalUn: Empowering Machine Unlearning via Gradient-based Weight Saliency in Both Image Classification and Generation. In The Twelfth International Conference on Learning Representations, 2024.
>
> [3] Jinghan Jia, Jiancheng Liu, Parikshit Ram, Yuguang Yao, Gaowen Liu, Yang Liu, Pranay Sharma, and Sĳia Liu. Model sparsity can simplify machine unlearning. Advances in Neural Information Processing Systems, 2023.
>
> [4] Min Chen, Weizhuo Gao, Gaoyang Liu, Kai Peng, and Chen Wang. Boundary unlearning: Rapid forgetting of deep networks via shifting the decision boundary. In Proceedings of the IEEE/CVF Conference on Computer Vision and Pattern Recognition, 2023.

---

> ### Author Response · Authors · 2025-11-17
>
> **W2. MIA-NN vs. MIA:** The reviewer is correct; MIA-NN is a distributional test that evaluates the success of unlearning a class from the model. The reason that we kept MIA in the name is that we are still evaluating the membership of a class in the dataset (rather than an individual sample) in the training set of the data. But we understand the reviewer’s point about the confusion it might cause to the reader. We propose changing the name to CMIA (class membership inference attack) to avoid confusion.
>
> **W3. Decision boundary geometry:** As the reviewer pointed out, we are introducing a new first-order constraint based on the similarity to the remaining classes when projecting the original distribution to the lower dimensional simplex. This has helped us to get a model that is more similar to the retrain models based on the existing metrics for evaluating the unlearning method, in addition to the new metric we introduce in this work. Because of the nature of the approximate unlearning methods, which unlike certified unlearning methods, are not accompanied by theoretical guarantees, the success is evaluated by the proposed methods. The current approach in the literature toward both problems, unlearning and evaluation, is the best-effort approach. Therefore, we have included all the existing SOTA methods and the most recent evaluation methods in our experiments.
>
> To further evaluate the effectiveness of our method in making the original model more similar to the retrain models, we conducted a study to evaluate the high-level geometry of the decision boundaries by looking at the class-wise clustering of samples embedding. The following link shows the t-SNE plot of the CIFAR-10 embeddings derived from a trained ResNet-18 model before and after unlearning as a comparison of how these embeddings look like in retrain models. As the figures show, in the original model, the classes are well-separated in the embedding space into distinct clusters. In the retrain model, where class 1 (automobile) has been removed from the training data, the clusters corresponding to class 1 (automobile) and class 10 (Truck) have been merged. The third figure, which shows that our method TRW effectively replicates this behavior.
>
> https://tinylink.net/7Ko0G
>
> As the reviewer mentioned, replicating the exact behavior of the retrained models might require higher-order constraints, but that would also add additional free hyper-parameters (only $\beta$ in the current model) and unknowns to the model, which require further assumptions about the underlying geometry of the decision boundaries. It is also possible to over-constrain the model or derive a feasible set that does not contain the retrain model at all. Therefore, we prioritize the more simple and interpretable model, which also shows good results based on the current evaluation metrics.

---

> ### Author Response · Authors · 2025-11-17
>
> **W4. Methodological details:**
>
> 1. Notice that $\sum_{y\neq f} q(y\mid x)\,s_y=c$ in the definition of set $\mathcal{A}$ is the expectation of $s_y$ values under the distribution $q(y\mid x)$. If we replace $q(y|x)$ with the rescaled distribution $\tilde p$ (equation 1), some value $c_0$ will be derived for this expectation. Now any value $c>c_0$ will lead to a set of feasible distributions in $\mathcal{A}$ that are more biased toward the more similar classes. This is because to achieve a higher expectation over the same $s_y$ values they need to assign higher probabilities to the larger $s_y$ values.
>
>     Now, as mentioned by the reviewer, in proposition 3.1 we show that minimizing the KL-divergence from the original distribution to this constrained set takes the form of equation (2). The form given in equation (2), replaces the hyper-parameter $c$ with $\beta$. The advantage of the new hyper-parameter is that setting that to $0$ will be equivalent to setting $c$ to $c_0$ and therefore any positive value for $\beta$ is equivalent to larger bias toward the more similar classes.
>
>     We have presented an ablation study on the value of $\beta$ in section B.7 which shows a tradeoff between accuracy of the model and effectiveness of unlearning as the value of $\beta$ changes. Our experiments showed that a value of $10$ for $\beta$ achieves effective unlearning without degrading the model's accuracy on the remaining classes ($Acc_r$).
>
> 2. We agree with the reviewer; we will add additional details after equation (2) to make it more clear.
>
> 3. Thank you for pointing this out. We removed the definition of TRW-2R from the main text due to space limitations but did not correctly move it to the appendix. The only difference between TRW-2R and TRW is that we randomly select two layers and fine-tune only their parameters using the loss function given in equation (4). The reported results are averaged over $10$ different random choices of these two layers. We proposed this variant for computational efficiency (as shown in Figure 5), while maintaining comparable performance. Moreover, selecting the two layers at random ensures that an adversary with access to the model cannot easily determine which layers were fine-tuned, adding an extra layer of security.
>
> **W5. Citation format:**  Thank you for the reminder, we changed cite{} to citep{} wherever necessary.

---

> ### Author Response · Authors · 2025-11-24
>
> We hope the reviewer has had an opportunity to review our responses. We believe we have addressed all of their concerns, including providing additional results from new experiments, and we hope the reviewer will consider increasing their score toward acceptance. If there are any further questions or concerns, we would be happy to continue the discussion.

---

> > ### Comment · Reviewer_HDYu · 2025-11-28
> > **Thank you for responses**
> >
> > Thank you for the detailed responses and clarifications.
> >
> > At this stage, I will keep my current score for this version of the paper. My main remaining concern is about scope and depth: the paper aims to do two substantial things at once, the MIA-NN and TRW. And as a result, each part feels somewhat not deep enough. It would better if we only dig one problem deeply in one paper.
> >
> > Also, note that ICLR allows authors to upload a revised, change-tracked manuscript during the rebuttal period. Instead of providing an external link, you can upload the updated version directly on OpenReview so that all reviewers and ACs can easily access the revision.

---

> > > ### Author Response · Authors · 2025-11-28
> > >
> > > Thank you for reviewing our responses! We will reflect all the new changes in our final revision. Thanks for raising questions that helped with further clarification in our methodology.
> > >
> > > We agree that MIA-NN and TRW are both substantial; our intent is for them to function as a diagnostic–remedy pair. MIA-NN exposes a specific failure mode (probability mass redistributing toward similar classes) in existing unlearning methods, which directly motivates TRW’s similarity-tilted objective. While each component can stand alone, presenting TRW without the underlying motivation and goal which is derived from the weakness revealed by MIA-NN raises concerns about the necessity of such tilting toward more similar classes.
> > >
> > > Regarding the reviewer’s concern about the depth of each contribution, we should note that we have already conducted several independent experiments for both MIA-NN and TRW. For TRW, we have already used **all other common evaluations that are used by prior class unlearning papers** using different models and datasets.
> > >
> > > To further deepen MIA-NN and address the reviewer’s concern, beyond the experiments already in the paper we made the following changes during the rebuttal period: (1) clarified the threat model; and (2) extended the attack to the black-box setting with additional experiments.

---

### Official Review · Reviewer_YnLx · 2025-10-25

**Soundness:** 3
**Presentation:** 1
**Contribution:** 2
**Rating:** 2
**Confidence:** 3

**Summary:**

This paper presents a new membership inference attack on the class unlearning task and then proposes a novel unlearning method with design robustness against membership inference attacks. The author made an observation that the retrained model demonstrates consistent misclassification of some retained classes when tested on the unlearning class. Based on the observation, the author designed MIA-NN, which exploits the probability assigned to the closest forget set to achieve a stronger MIA. The paper proposes Tilted Re-Weighting(TRW) to achieve secure unlearning under MIA. TRW redistributes the probability among the remaining classes to achieve a consistent prediction with a retrained model.

**Strengths:**

1. The proposed unlearning method is clear and insightful.
2. Strong theory foundation with clear mathematical explanation.
3. Comprehensive benchmark method comparison.

**Weaknesses:**

1. The table description is inaccurate and causes confusion due to the lack of notation in the table. Specifically, Table 1 is described as follows: " Higher values indicate better unlearning; however, the paper also describes that the gap between the Acc_i and Acc^Mu_rn is used to measure the unlearning effectiveness. In the experiment section, the paper proposes that the MIA score of the unlearned model is expected to match that of the retraining model. However, the table does not provide the MIA score of any retraining model or the difference between the unlearned model and the retrained model. These problems pose difficulties in understanding the results of the paper.
2. In Table 2 and Table 4, the method TRW-2R outperformed TRW unlearning  introduced in the paper. In Appendix B.5, the author also mentioned the proposed TRW-2R as the fastest among other baseline methods. However, there's a lack of the implementation and theoretical details of the methodTRW-2R.  The authors do not explicitly specify what additional components or improvements has been made to distinguish TRW-2R from TRW. Without the details, it’s hard to understand and verify the reason for the experiment improvement.
3. The discussion of the method is limited to class unlearning on classical vision models. This presents a dual limitation for extending this method to either other types of models(for example, GNN and LLM) or type of unlearning (for example, per-example unlearning). I would appreciate some discussion of the possibility of extending this method to other models, such as ViT.

**Questions:**

One of the important insights from this paper is the vulnerability of the unlearned model after class unlearning, and a corresponding attack method, MIA-NN, has been proposed to exploit it. I am curious why MIA-NN is not part of the evaluation matrix(except for B.7 with an ablation study focus on the effect of beta).

---

> ### Author Response · Authors · 2025-11-17
>
> We thank the reviewer for their insightful comments. We are glad that they found our method to be insightful and theoretically well-founded. Below are our responses to their questions and corresponding weaknesses:
>
> **W1. Clarifying the table descriptions:** Thanks for pointing out the potential confusion due to using multiple different metrics. Similar to prior works [3] in class unlearning, our goal is to get as close as possible to the retrained models. In Table 1 the results for the retrain models are given in the column Retrain, and since the values for other methods were smaller than the one from the retrain model, we have mentioned the goal as achieving a score as high as the retrain model, but the goal is to minimize the gap; we will revise the wording in the caption to reflect this. TRW attains the performance most closely matching the Retrain models, with a gap of $7.5\%$ on CIFAR-10’s _automobile_ class, representing a $54.0\%$ improvement over the best result from prior methods.
>
> Other than MIA-NN, we have used other metrics used in prior class unlearning works, including the gap of the unlearned model in terms of $Acc_r$ (accuracy on the retrained model), $Acc_f$ (accuracy of the forget samples), and regular MIA [1,2,3]. For these metrics the goal is to achieve high $Acc_r$ while minimizing $Acc_f$ and achieve a MIA score of $100\%$ similar to the retrained models. The results for the Retrain model are given in the row for Retrain in Table 2 of our paper.
>
> To further clarify these results and decrease confusion, we will add the MIA-NN results (Table 1) as a new column to Table 2 and provide a column called “Avg Gap” that shows the average gap of all the metrics ($Acc_r$,  $Acc_f$, MIA, and MIA-NN) with the scores of the Retrain models for further clarity.
>
> In addition to these metrics, we also used another recently proposed metric U-LiRA (Table 3) which is specifically designed for evaluating the unlearning methods. For this metric, the ideal result is $50\%$, the same score that is expected from the Retrain models (as explained in the caption). We will add a specific column for the Retrain model to further clarify this table.
>
> **W2. TRW-2R:** Thank you for pointing this out. We removed the definition of TRW-2R from the main text due to space limitations but did not correctly move it to the appendix. The only difference between TRW-2R and TRW is that we randomly select two layers and fine-tune only their parameters using the loss function given in equation (4). The reported results are averaged over $10$ different random choices of these two layers. We proposed this variant for computational efficiency (as shown in Figure 5), while maintaining comparable performance. Moreover, selecting the two layers at random ensures that an adversary with access to the model cannot easily determine which layers were fine-tuned, adding an extra layer of security.

---

> ### Author Response · Authors · 2025-11-17
>
> **W3. Extension to other models:** Because of the differences in the nature of model training, objective, and definition of unlearning in these domains, LLMs, GNNs, and classification models have different approaches to unlearning. Most prior works in class unlearning for classification models do not extend to the setting of LLMs and GNNs [1,2]. That being said, a few prior works in unlearning for classification models extend their approach to the setting of text-to-image generative models [3]. It is interesting to note that our method can be extended to the setting of text-to-image generative models. Consider a generative model (e.g., diffusion model) that has learned the conditional distribution $p_\theta(x | c_f)$ for the forget class/concept $c_f$. To unlearn from this model, we would be able to set the tilted distribution as follows:
>
> $q^*(x | c_f) \propto \sum_{c \in \mathcal{C_r}} \exp(\beta s(c, c_f))  p_\theta(x | c) $
>
> As our target distribution, where $s(c,c_f)$ measures semantic similarity (e.g., CLIP embeddings) between the forget concept $c_f$ and the remaining concepts $c \in \mathcal{C}_r$ and $\beta$ controls the tilt. Using this tilted distribution as the target distribution would bias the model toward concepts that are more semantically similar to $c_f$, leading to utility preservation of the model while unlearning class $c_f$. Similar approach could be used for unlearning a node in GNNs by considering the more similar neighbors. In general, our framework is very flexible and various similarity measures could be used to derive the desired form of the target distribution. We will add a discussion of this idea in our future work section and leave the implementation details to future work.
>
> [1] Jinghan Jia, Jiancheng Liu, Parikshit Ram, Yuguang Yao, Gaowen Liu, Yang Liu, Pranay Sharma, and Sĳia Liu. Model sparsity can simplify machine unlearning. Advances in Neural Information Processing Systems, 2023.
>
> [2] Min Chen, Weizhuo Gao, Gaoyang Liu, Kai Peng, and Chen Wang. Boundary unlearning: Rapid forgetting of deep networks via shifting the decision boundary. In Proceedings of the IEEE/CVF Conference on Computer Vision and Pattern Recognition, 2023.
>
> [3] Fan, C., Liu, J., Zhang, Y., Wong, E., Wei, D., \& Liu, S. (2023, October). SalUn: Empowering Machine Unlearning via Gradient-based Weight Saliency in Both Image Classification and Generation. In The Twelfth International Conference on Learning Representations, 2024.
>
> **Q1. MIA-NN results:** The results of MIA-NN for our method and all the baselines were presented in Table 1 of our paper, but for further clarification we added it as a new column to Table 2 (averaged over 10 of the classes in each dataset) and added an “Avg Gap” column that shows the average gap of all the metrics ($Acc_r$,  $Acc_f$, MIA, and MIA-NN) with the scores of the Retrain models for further clarity:
>
> https://tinylink.net/MEpZb

---

> ### Author Response · Authors · 2025-11-24
>
> We hope the reviewer has had an opportunity to review our responses. We believe we have provided responses to all the reviewer’s concerns and we hope the reviewer considers increasing their score toward acceptance. If there are further questions and concerns, we are happy to continue to engage.

---

> ### Comment · Reviewer_YnLx · 2025-11-28
>
> Dear Authors,
>
> Thank you for responding to my comments!  My concerns are addressed with one small caveat.  I wanted to raise my score, but the system will not let me do so.  (I cannot edit my review for some reason.) But for the Area Chair (WkpA), my concerns are well addressed.
>
> Here are additional notes:
>
> W1:Thank you for the planned update and clarification. I appreciate the precise and consistent terminology used
> throughout the paper.
>
> W3: Thank you for this response. I believe the authors provide a clear and sufficient discussion regarding how the
> method may be extended to other models. This addresses my concern on weakness 3.
>
> Q1: Thank you for the updated table including MIA-NN as a new column. This makes the tables much easier to read
> and interpret.
>
> W2: It would make the paper stronger if you explained TRW-2R in the caption.

---

### Official Review · Reviewer_QFzx · 2025-10-30

**Soundness:** 3
**Presentation:** 3
**Contribution:** 3
**Rating:** 6
**Confidence:** 4

**Summary:**

The paper offers a crisp diagnostic (MIA-NN) and a pragmatic fix (TRW) that together materially advance evaluation and practice of class unlearning. While the core ideas are simple, they are impactful and well-validated. Clarifications on the attack’s practicality and broader stress-testing would further solidify the case for acceptance.

**Strengths:**

1. The paper pinpoints a blind spot in current class-unlearning evaluation—evaluations ignore how retrained models systematically misclassify forgotten-class samples toward semantically similar retained classes. This motivates a new attack (MIA-NN) that probes leakage via the nearest neighbor of the forgotten class and exposes failures of many SOTA methods.
2. The proposed Tilted ReWeighting (TRW) modifies the fine-tuning objective by zeroing the forget label and tilting the remaining class distribution using inter-class similarities derived from logit weights; the resulting target distribution can be seen as an information projection with a linear moment constraint. It is lightweight (drop-in during fine-tuning) and conceptually clean.

**Weaknesses:**

1. TRW hinges on a particular class-similarity score (cosine in a PCA-projected logit space with a sharp softmax temperature). While intuitive, this is heuristic and sample-independent; performance sensitivity to the choice of similarity, PCA dimension, temperature, and $\beta$ needs deeper analysis beyond brief ablations.
2. The attack identifies a “nearest neighbor” via statistics from multiple retrained models and trains an SVM on logits. Although the paper claims the attack does not assume access to training data, the practicality of assembling enough scratch-retrained references (and the knowledge required) deserves clearer discussion and a black-box-only variant analysis.

**Questions:**

1. How many scratch-retrained models are required for MIA-NN to be reliable, and under what access (black-box probabilities only vs. logits vs. labels)? Can you report MIA-NN performance under strictly black-box probability queries and with one or zero reference retrains (e.g., using public checkpoints as surrogates)?
2. How sensitive is TRW to (i) PCA dimension, (ii) softmax temperature over similarities, (iii) cosine vs. centroid-distance vs. feature-space CKA similarities, and (iv) the tilt parameter $\beta$  (beyond the brief ablation)? Please include a grid showing ACCr/ACCf/MIA/U-LiRA vs. these hyperparameters.

---

> ### Author Response · Authors · 2025-11-17
>
> We thank the reviewer for their insightful comments. We are excited about the reviewer’s acknowledgment of the usefulness of our new evaluation metric (MIA-NN) and conceptual cleanness of our proposed approach (TRW). Below are our responses to their questions and corresponding weaknesses:
>
> **Q1+W2. MIA-NN:** Based on our experiments, one retrained model is enough for evaluations as the accuracy values of the SVM classifier that we computed on the retrained models have very low standard deviation. In the updated version of Table 1 available via the following link, we have added the standard deviations for 3 different random seeds:
>
> https://tinylink.net/hINCs
>
> Although the retrain models used by the adversary have been trained on the training set (which is the common practice in the evaluation of unlearning methods [1,2,3]), we can drop this assumption and utilize the models retrained by the adversary on public data (disjoint from the training set).
>
> To perform this experiment and show-case the performance of MIA-NN in a black-box setting, we randomly split the CIFAR-10 training set into two parts. We use only the first part for training the original model. The adversary has access to the second part, which is disjoint from the training samples and can be considered public data in this setting, and uses that for training one retrained model. Then the adversary utilizes the CIFAR-10 test set (which is again considered public data) and the output probabilities from the original model to evaluate whether the original model has been trained on the forgotten class. Therefore, in this setting, we have shown the use-case of MIA-NN in a black-box setting. The following table, shows the results in the black-box setting for unlearning one of the CIFAR-10 classes from a trained ResNet18 model. The results are also consistent with the ones we have in table 1 of our paper.
>
> https://tinylink.net/v1fMv
>
> If a public checkpoint for a model that has been trained on only the remaining classes is available to the adversary, the adversary would be able to utilize that model as it provides a similar setting to the one we used for our black-box setting (no access to the training data and the original model parameters). Still, this new black-box experiment, which requires only one retrained model on public data to reveal the shortcomings of existing unlearning methods is a rigorous and realistic threat model that also provides the community with a useful metric for evaluations of class unlearning methods.
>
>
> [1] Fan, C., Liu, J., Zhang, Y., Wong, E., Wei, D., \& Liu, S. (2023, October). SalUn: Empowering Machine Unlearning via Gradient-based Weight Saliency in Both Image Classification and Generation. In The Twelfth International Conference on Learning Representations, 2024.
>
> [2] Jinghan Jia, Jiancheng Liu, Parikshit Ram, Yuguang Yao, Gaowen Liu, Yang Liu, Pranay Sharma, and Sĳia Liu. Model sparsity can simplify machine unlearning. Advances in Neural Information Processing Systems, 2023.
>
> [3] Min Chen, Weizhuo Gao, Gaoyang Liu, Kai Peng, and Chen Wang. Boundary unlearning: Rapid forgetting of deep networks via shifting the decision boundary. In Proceedings of the IEEE/CVF Conference on Computer Vision and Pattern Recognition, 2023.

---

> ### Author Response · Authors · 2025-11-17
>
> **Q2+W1. Hyper-parameters in TRW:** In addition to the ablation study on the tilt parameter ($\beta$) in section B.7 of our paper, we performed an ablation study on the other hyper-parameters of our method and they are presented in the following table:
>
> https://tinylink.net/4uHsZ
>
>
> Our experiments showed that the use of cosine similarity of class-wise weight vectors in the last layer (82.1 MIA-NN) leads to empirically stronger and closer to the retrain ideal (90.1) compared to the Euclidean distances (77.8). It also shows that our model is robust to slight changes in PCA dimension, tilt parameter ($\beta$), and inverse temperature in computing the similarity scores. Although we fixed our hyper-parameters in all our experiments (different models, datasets, and forget classes) to: a) cosine similarity of the class-wise weight vectors, b) PCA dimension of $32$, c) inverse temperature of $5$, and d) $\beta=10$, as the results show, hyper-parameter tuning can lead to slight improvements in some cases.
>
> In addition to this regular ablation study, you can find the results of another experiment we conducted to show the effect of inverse temperature (for computing the similarity scores) and the tilt parameter ($\beta$) in our method when applied to unlearning the _automobile_ class from a ResNet-18 model trained on CIFAR-10. In this experiment, we are interested in finding out how the weight given to the nearest-neighbor class (_truck_) in equation 2 of our paper changes. Basically, we compute $\frac{\exp(\beta\,s_y)}{\sum_{j\neq y_f}\exp(\beta\,s_j)}$ for the _truck_ class. Note that for the rescaled distribution given in equation 1 (which is equivalent to setting $\beta = 0$ in equation 2), this value is equal to $\frac{1}{9} \approx 0.11$. As the figure in the link shows, for the default parameters used in our experiments ($\beta = 10$ and an inverse temperature of $5$), this leads to a modest boost in the weight of the most similar class ($0.15$). This modest boost is due to the fact that the cosine similarity values that we compute for the class-wise weight vectors lead to very small differences for some of the classes. For example, for the _automobile_ class, the cosine similarities with the three most similar classes _truck_, _ship_, and _airplane_ are $0.937$, $0.922$, and $0.919$, accordingly. Therefore, as shown in the figure and the presented table, our method is robust to modest changes in the value of $\beta$ and the inverse temperature and only deteriorates at extreme values.
>
> Also note that a benefit of using the initial softmax for computing the similarities is that despite the choice of the similarity metric and the range of values, it converts the similarity values to probabilities in range $[0,1]$ and therefore prevents extreme changes from appearing when computing the weights in the tilted distribution of equation 2.

---

> ### Author Response · Authors · 2025-11-24
>
> We hope the reviewer has had an opportunity to review our responses! Because of encouraging comments from the reviewer about our work and and our efforts to address the raised concerns through additional experiments, we would appreciate it if the reviewer could consider increasing their score to send a clear signal to the AC. If there are further questions and concerns, we are happy to continue to engage.

---

### Meta-Review · Area_Chair_6Wbh · 2026-01-07

**Summary:**

Based on the three reviews provided, the feedback focused on three major areas: methodological clarity, experimental rigor, and the definition/practicality of the proposed attack (MIA-NN).  The appearance of a method called TRW-2R in results tables that is never defined in the text. There is a lack of theoretical or empirical justification for the choice of the parameters.  The goals, capabilities, and roles of the attacker are not clearly defined in the text. Limited discussion on extending the framework to Vision Transformers (ViTs) or other architectures.

**Reviewer Concerns:**

all reviewers have raised issues on clarity of the paper, including missing definitions of key concepts, missing threat model, confusing notations etc. Although rebuttals are provided to address these issues, they require substantial changes to the original paper which would need another round of careful review before it can be accepted.

**Reviewer Scores:**

Reviewer YnLx would raise score based on his/her feedback.
Reviewer HDYu would remind his/her score based on his/her feedback.
Reviewer QFzx would probably maintain or raise his score. No further feedback is provided from this reviewer.

---

### Decision · Program_Chairs · 2026-01-26

Reject